# *Mycobacterium tuberculosis* H_2_S Functions as a Sink to Modulate Central Metabolism, Bioenergetics, and Drug Susceptibility

**DOI:** 10.3390/antiox10081285

**Published:** 2021-08-13

**Authors:** Tafara T. R. Kunota, Md. Aejazur Rahman, Barry E. Truebody, Jared S. Mackenzie, Vikram Saini, Dirk A. Lamprecht, John H. Adamson, Ritesh R. Sevalkar, Jack R. Lancaster, Michael Berney, Joel N. Glasgow, Adrie J. C. Steyn

**Affiliations:** 1Africa Health Research Institute, University of KwaZulu Natal, Durban 4001, South Africa; tafara.kunota@ahri.org (T.T.R.K.); aejaz.rahman@ahri.org (M.A.R.); barry.truebody@ahri.org (B.E.T.); jared.mackenzie@ahri.org (J.S.M.); DLamprec@its.jnj.com (D.A.L.); john.adamson@ahri.org (J.H.A.); 2Department of Biotechnology, All India Institute of Medical Sciences, New Delhi 110029, India; vikram@aiims.edu; 3Department of Microbiology, University of Alabama at Birmingham, Birmingham, AL 35294, USA; ritesh09@uab.edu (R.R.S.); jng@uab.edu (J.N.G.); 4Department of Pharmacology and Chemical Biology, Vascular Medicine Institute, University of Pittsburgh School of Medicine, Pittsburgh, PA 15261, USA; doctorno@pitt.edu; 5Department of Microbiology and Immunology, Albert Einstein College of Medicine, New York, NY 10462, USA; michael.berney@einsteinmed.org; 6Centers for AIDS Research and Free Radical Biology, University of Alabama at Birmingham, Birmingham, AL 35294, USA

**Keywords:** *Mycobacterium tuberculosis*, H_2_S, energy metabolism, bioenergetics, respiration, metabolomics, redox homeostasis, ergothioneine, mycothiol, CytBD, cysteine, Rv3684, Agilent Seahorse XFe96

## Abstract

H_2_S is a potent gasotransmitter in eukaryotes and bacteria. Host-derived H_2_S has been shown to profoundly alter *M. tuberculosis* (*Mtb*) energy metabolism and growth. However, compelling evidence for endogenous production of H_2_S and its role in *Mtb* physiology is lacking. We show that multidrug-resistant and drug-susceptible clinical *Mtb* strains produce H_2_S, whereas H_2_S production in non-pathogenic *M. smegmatis* is barely detectable. We identified Rv3684 (Cds1) as an H_2_S-producing enzyme in *Mtb* and show that *cds1* disruption reduces, but does not eliminate, H_2_S production, suggesting the involvement of multiple genes in H_2_S production. We identified endogenous H_2_S to be an effector molecule that maintains bioenergetic homeostasis by stimulating respiration primarily via cytochrome *bd*. Importantly, H_2_S plays a key role in central metabolism by modulating the balance between oxidative phosphorylation and glycolysis, and it functions as a sink to recycle sulfur atoms back to cysteine to maintain sulfur homeostasis. Lastly, *Mtb*-generated H_2_S regulates redox homeostasis and susceptibility to anti-TB drugs clofazimine and rifampicin. These findings reveal previously unknown facets of *Mtb* physiology and have implications for routine laboratory culturing, understanding drug susceptibility, and improved diagnostics.

## 1. Introduction

Tuberculosis (TB) remains a major cause of morbidity and mortality in underdeveloped and developing countries. While substantial progress has been made in understanding the biological basis of *Mtb* pathogenesis, precisely how *Mtb* physiology and metabolism contribute to persistence, pathogenesis, and drug resistance is poorly understood.

The gasotransmitters carbon monoxide (CO) and nitric oxide (NO) have important roles in *Mtb* physiology [1,2,3,4]. A third gasotransmitter, hydrogen sulfide (H_2_S), is involved in a wide variety of physiological processes in eukaryotes and prokaryotes [5]. H_2_S is a weak acid in solution and at physiological pH, it exists predominantly as a hydrosulfide anion (HS^−^, 75–80%), with the rest as H_2_S (20–25%) and only trace amounts of S^2−^ [6]. Notably, H_2_S diffuses easily through cell membranes unlike its deprotonated conjugate bases HS^−^ and S^2−^. The complex nature of H_2_S in solution and its volatility make this molecule very difficult to measure and challenging to work with [6,7]. Furthermore, H_2_S can be generated non-enzymatically from media components [8] and, therefore, methods for measuring H_2_S require rigorous validation.

Cysteine (Cys) is frequently used as a substrate by bacterial Cys desulfhydrases for H_2_S production. Intracellular Cys levels require careful control, since excessive levels can stimulate oxidative stress via the Fenton reaction, which produces hydroxyl radicals [9]. On the other hand, H_2_S is also a substrate for Cys biosynthesis, which is required for the formation of low molecular weight thiols such as mycothiol (MSH) and ergothioneine (EGT) and biogenesis of iron sulfur cluster proteins [10,11]. The ability of bacteria to produce H_2_S has been widely used as a diagnostic test and for taxonomic purposes [12,13].

H_2_S production in bacteria has been attributed mainly to the enzymatic activity of cystathionine β-synthase (CBS), cystathionine γ-lyase (CSE), and 3-mercaptopyruvate sulfurtransferase (3MST) [14,15]. Addition of Cys to bacterial cultures has been shown to stimulate H_2_S production [14,15,16,17]. In both bacterial and mammalian cells, Cys is used as a substrate by CBS and CSE to produce H_2_S [15,18,19]. Although bacterial production of H_2_S was initially considered to be a byproduct of sulfur metabolism with unclear physiological relevance, studies in several bacterial species have shown that disrupting H_2_S-producing genes increases antibiotic susceptibility [15]. Obviously, these findings have important implications for understanding drug resistance in bacterial pathogens. In *E. coli*, 3MST produces the majority of cellular H_2_S from Cys [14] and genetic disruption of this gene leads to increased sensitivity to oxidative stress [14], implicating 3MST-derived H_2_S in maintaining redox homeostasis.

Several studies have shown that H_2_S reversibly inhibits cytochrome c oxidase (Complex IV) at high concentrations and stimulates mitochondrial respiration at low concentrations [20,21]. We have recently shown that exogenous H_2_S also targets the *Mtb* electron transport chain to increase respiration and ATP levels leading to increased growth [22], and that host-generated H_2_S exacerbates TB in mice [22,23]. Further, CSE and 3MST protein levels are substantially increased in human tuberculous lung tissue, and it has been proposed that *Mtb* triggers supraphysiological levels of host-generated H_2_S at the site of infection to suppress host immunity, thereby exacerbating disease [23]. Notably, *Mtb* senses host-generated H_2_S during infection and reprograms its metabolism accordingly [22].

H_2_S plays important roles in modulating mammalian immunity, although its effects have been shown to be both pro- and anti-inflammatory [18,24]. Thus, H_2_S produced by bacterial pathogens could act as a signaling molecule in the host, potentially exacerbating disease. The presence of *Mtb* homologues of H_2_S-producing enzymes identified in KEGG [25] and several biochemical studies [26,27,28,29,30] suggest that *Mtb* has the capacity to produce H_2_S. However, formal genetic and biochemical demonstration of H_2_S production by *Mtb* cells and its functions in *Mtb* physiology is lacking. Due to the diverse roles H_2_S plays in prokaryotic biology and bacterial classification, clear evidence that *Mtb* produces H_2_S is expected to widely influence our understanding of *Mtb* physiology, disease, and diagnostics.

Here, we tested the hypothesis that *Mtb* produces H_2_S by measuring H_2_S production in drug-susceptible (DS) and multidrug-resistant (MDR) clinical strains, laboratory *Mtb* strains, and non-pathogenic mycobacterial species. We then identified a gene and corresponding protein responsible for H_2_S production in *Mtb*. We generated an *Mtb* knockout strain and utilized flow cytometry, extracellular metabolic flux analysis, ^13^C stable isotope analyses, and drug studies to determine the effect of endogenously produced H_2_S on respiration, central metabolism, redox balance, and drug susceptibility. We expect our findings to broadly impact our understanding of *Mtb* physiology and drug resistance.

## 2. Materials and Methods

### 2.1. General

All mycobacteria strains (See Appendix A) were cultured in Middlebrook 7H9 media (BD Difco, New York, NY, USA) supplemented with 0.01% tyloxapol (Sigma–Aldrich, USA), 0.2% glycerol (Sigma–Aldrich, New York, NY, USA), and 10% (oleic acid, bovine albumin fraction V, dextrose, and catalase (OADC, BD Difco, New York, NY, USA) unless stated otherwise. Cultures were placed in a shaking incubator (100 rpm) at 37 °C. Strains examined included *Mb* H37Rv, *Mtb* CDC1551, *Mycobacterium bovis* (supplemented media with 100 µM sodium pyruvate), *Mycobacterium bovis* BCG, *M. smegmatis*, two drug susceptible *Mtb* strains (i.e., TKK-01-0027 and TKK-01-0047), and two multi-drug resistant *Mtb* strains (i.e., TKK-01-0035 and TKK-01-0001). The drugs’ MIC_50_ values used during this study were as follows: clofazimine (CFZ), 211 nM; rifampicin (RIF), 486 nM; isoniazid (INH), 240 nM. Where required, the following antibiotics were used; hygromycin B (100 µg/mL for *E. coli*, 50 µg/mL for mycobacteria) and kanamycin (50 µg/mL for *E. coli*, 25 µg/mL for mycobacteria). Dihydroethidium (DHE) was purchased from Thermo Fisher Scientific (New York, NY, USA) (Cat# D11347). Restriction enzymes were obtained from Thermo Fisher Scientific (Germany). The KOD Xtreme Hotstart DNA polymerase kit was obtained from Merck (Darmstadt, Germany). T4 DNA ligase was obtained from New England Biolabs (NEB, New York, NY, USA). *E. coli* DH5α, used for cloning and DNA manipulation, was routinely cultured in Luria-Bertani liquid media at 37 °C. Oligonucleotides were synthesized by Thermo Fisher Scientific (USA). All other reagents were purchased from Merck or Sigma–Aldrich.

### 2.2. Preparation of Mycobacterial Lysates

All cultures were grown to an OD_600_ of ~0.8. The cells were then harvested from 30 mL of culture and centrifuged at 4000× *g* for 5 min. The supernatant was discarded, and the pellet was resuspended in 1–2 mL of lysis buffer (50 mM Tris-HCl, pH 8.0; 150 mM NaCl; protease inhibitor (Roche, New York, NY, USA)), depending on the size of the cell pellet. Cells were lysed in a MagNA Lyser (Roche, USA) at 7000 rpm for 1 min and then placed on ice for 4 min. This was repeated 3–4 times. The lysates were then centrifuged at 15,000× *g* for 10 min. The supernatant was collected and passed through a 0.22 µm filter. Protein concentrations were determined using the Micro BCA Protein Assay Kit (Thermo Fisher Scientific, USA), and the absorbance at 562 nm was measured using a Biotek Synergy H4 Hybrid Reader (BioTek, New York, NY, USA. Lysates were stored at 80 °C until use.

### 2.3. H_2_S Measurement Using the Lead Acetate Assay

Mycobacterial cultures were harvested at an OD_600_ of 0.8–1 and centrifuged at 3500× *g* for 5 min. The bacterial pellet was resuspended in an equal volume of 7H9 media containing 0.01% tyloxapol, 0.2% glycerol, and 10% OADC. 10 mL of diluted culture at an OD_600_ of 0.1 was then transferred to a 30 mL culture bottle. Lead acetate strips (Thermo Fisher Scientific, USA) were affixed to the inner wall of the culture bottles. The strips were monitored for the formation of dark colored lead sulfide precipitate and scanned after 48 h. The intensity of the dark lead sulfide stain is proportional to the amount of H_2_S present. The lead sulfide stain was then scanned and quantified by measuring the grayscale values for a specific area of each strip and normalized to OD_600_ using ImageJ software version 1.53a (Java 1.8.8_12 (64 bit)) [31].

### 2.4. H_2_S Measurement Using the Bismuth (III) Chloride (BiCl_3_) Assay

The BiCl_3_ (BC) assay is used to measure H_2_S based on the reaction of H_2_S with a bismuth (III) salt to form bismuth (III) sulfide (Bi_2_S_3_), which appears as a brown-to-black precipitate [32]. The microplate BC assay was performed in 96-well plates using intact H37Rv bacteria and lysates as described by Basic et al. (2015) [33]. Once the OD_600_ of cultures reached ~0.8–1, the cultures were centrifuged at 3500× *g* for 5 min. The supernatant was discarded, and the culture pellet was resuspended in the original volume of media before centrifugation. The BC assay solution (2×) contains 0.4 M triethanolamine–HCl/Tris–HCl (Sigma–Aldrich, USA), pH 8.0; 1 mM BiCl_3_ (Sigma–Aldrich, New York, NY, USA); 20 µM pyridoxal 5-phosphate monohydrate (PLP) (Sigma–Aldrich, USA), 20 mM EDTA (Sigma–Aldrich, USA), and 40 mM l-cysteine (Cys) (Sigma–Aldrich, USA). One hundred microliters of the mycobacteria cell suspension (OD_600_ = 1.0) or lysate (5 μg), with or without the inhibitors AOAA and PAG, was mixed with 100 µL of freshly prepared 2× bismuth solution in clear, flat-bottomed 96-well microtiter plates (Corning Inc., New York, NY, USA). For H_2_S measurements when *Mtb* was grown on different carbon sources, 100 µL of mycobacterial cell suspension in 7H9 medium containing 0.01% tyloxapol was supplemented with either: 0.4% glycerol, 1 mM sodium butyrate (Sigma–Aldrich, USA), 8 mM sodium acetate (Sigma), 2 mM sodium propionate (Sigma–Aldrich, USA) or 0.02% (*w*/*v*) cholesterol (Sigma–Aldrich, USA). These cell suspensions were then mixed with 100 µL of freshly prepared 2× bismuth solution in clear flat-bottomed 96-well microtiter plates. The working stock cholesterol was initially dissolved at 100 mg/mL in a solution of tyloxapol:ethanol (1:1) at 80 °C. Bi_2_S_3_ formation was determined by measuring the absorbance at 405 nm. The kinetics for mycobacterial cells was measured every 30 min for 15–20 h at 37 °C with shaking using a Hidex Sense Plate reader (Hidex, Finland). Enzymatic kinetics using lysates was measured every 5 min for 5–20 h with shaking at room temperature (~20–22 °C) using a Biotek Synergy H4 Hybrid Reader, (BioTek, New York, NY, USA).

### 2.5. Mtb Growth in Fatty Acids or Cholesterol as the Sole Carbon Source

One hundred microliters of mycobacterial cell suspension of OD_600_ ~0.1 in 7H9 containing 0.01% tyloxapol were mixed with 100 µL of 7H9 media containing 0.01% tyloxapol supplemented with either: 0.4% glycerol, 1 mM sodium butyrate, 8 mM sodium acetate, 2 mM sodium propionate, or 0.02% (*w*/*v*) cholesterol in clear, flat-bottomed 96-well microtiter plates. The plates were placed in an incubator at 37 °C. OD_600_ measurements were taken after 7 days.

### 2.6. H_2_S Measurement Using the Unisense Amperometric Microsensor

H_2_S released by cell cultures and lysates was measured at room temperature with a sensitive sulfide amperometric microsensor, H_2_S-500 (Unisense A/S, Denmark), connected to a microsensor multimeter (Unisense, A/S, Denmark) as an amplifier for data acquisition. The signal for H_2_S was collected in mV and converted to µM using a NaHS (Thermo Fisher Scientific) standard curve generated from a concentration range of 0–100 μM (freshly prepared in an anaerobic glovebox). The H_2_S microsensor was calibrated in accordance with the manufacturer’s instructions. Bacteria at an OD_600_ of 0.2 were cultured in media with or without 1 mM Cys. After 72 h, H_2_S concentrations were measured using the microsensor in the cell culture and the cell-free culture supernatants (referred to as “cleared supernatants”). Alternatively, *Mtb* strains were cultured to OD_600_ of ~0.8 without L-Cys and H_2_S measured after mixing 700 µL *Mtb* culture and 300 µL assay buffer (4 mL of 1.0 M Tris-HCl, pH 7.0, 1.0 mL of 400 mM L-Cys, and 1.0 mL of 200 mM EDTA) using the microsensor at different time points. H_2_S levels were normalized according to optical density.

For real-time H_2_S measurement in lysates, the microsensor was placed in a 2 mL tube containing 200 µL of 2× assay buffer (0.4 M triethanolamine-HCl, pH 8.0; 20 µM PLP, 20 mM EDTA) with or without 40 mM L-Cys and 160 µL lysis buffer (50 mM Tris-HCl, pH 8.0; 150 mM NaCl). The signal was allowed to generate a stable buffer baseline for ~5 min after which 40 µg (adjusted volume to 40 µL) of mycobacterial lysate was added to the reaction. When appropriate, L-Cys (0.1–4 mM) was subsequently added to the tube at different time intervals. To confirm AOAA inhibition, *Mtb* lysate preincubated with AOAA (4 mM) was added to the assay buffer containing 20 mM L-Cys. OASS activity was measured by placing the microsensor in 1 mL PBS solution. A 10 µL aliquot of 25 mM NaHS was added to the reaction tube twice, and after the signal stabilized, 30 ng of OASS was added to the reaction. After 2 min, OAS was added to the reaction to a final concentration of 10 mM, and the H_2_S signal was monitored in real time.

### 2.7. Native PAGE Analysis and In-Gel BC Assay

Equal amounts (15–25 µg per lane) of mycobacterial lysate were resolved on 10% PAGE gels (Bio-Rad, New York, NY, USA) under non-denaturing conditions using running buffer containing 25 mM Tris-base and 190 mM glycine. To detect the presence of H_2_S-producing proteins, gels were incubated in 20–50 mL of BC solution and incubated at room temperature with shaking. Gels were monitored every 20–60 min for the appearance of dark-colored Bi_2_S_3_. For gels exposed to AOAA, gels were incubated in 20 mL of 2 mM AOAA in 50 mM Tris-HCl pH 8.0 with shaking at room temperature for 5 min, followed by BC solution containing 2 mM AOAA overnight.

### 2.8. Extracellular Flux Analysis

The oxygen consumption rates (OCR) of *Mtb* strains were measured using a Seahorse XFe96 Extracellular Flux Analyzer (Agilent Technologies Inc., New York, NY, USA). *Mtb* bacilli were adhered to the bottom of a Cell-Tak-coated XF cell culture microplate at 2 × 10^6^ bacilli per well. Cell-Tak has no effect on *Mtb* basal respiration [34]. Assays were carried out in unbuffered 7H9 media (pH 7.35) with no carbon source. *Mtb* bacilli were grown in this unbuffered 7H9 media, containing only 0.01% Tyloxapol, for 24 h before being seeded into the XF cell culture microplate at the start of the experiment. In general, basal OCR was measured for ~25 min before automatic sequential injection of various compounds through the drug ports of the sensor cartridge. The duration of OCR measurements after compound addition and the concentrations used varied by experiment. OASS modulation of *Mtb* OCR in the presence of L-Cys was performed by the simultaneous addition of Cys, OASS, and substrate OAS (final concentration of 4 mM, 0.03 µg/mL, and 4 mM, respectively). Q203-based modulation of the OCR in *Mtb* and Δ*cydAB* cells was performed in the presence of the indicated Cys concentration followed by the addition of Q203 (300 × MIC_50_; MIC_50_ for Q203 is 3 nM) [34]. To chemically complement Δ*cds1* cells with H_2_S, different concentrations of NaHS were added to cells after measuring the baseline OCR. All OCR data figures indicate the time of each addition as dotted lines. OCR data points are representative of the average OCR during 4 min of continuous measurement in the transient microchamber, with the error being calculated from the OCR measurements taken from at least three replicate wells by the Seahorse Wave Desktop 2.2 software (Agilent Technologies Inc., New York, NY, USA). The transient microchamber was automatically re-equilibrated between measurements through the up and down mixing of the probes of the XF96 sensor cartridge in the wells of the XF cell culture microplate.

### 2.9. CFU-Based Assay

Mid-log phase mycobacterial cultures were diluted to an OD_600_ of 0.01 in 7H9 media. For survival studies in the presence of NaHS, bacterial cultures (7H9 with 10% OAD) were untreated or treated with anti-TB drugs and NaHS at indicated concentrations. For survival studies in the presence of antioxidants, bacterial cultures (7H9 with 10% OAD) were treated with or without 0.25 mM cumene hydroperoxide (CHP) for 16 h. For survival studies in the presence of CFZ, bacterial cultures (7H9 with 10% OADC) were treated with or without clofazimine at MIC 60×, 100× and 300× for 8 days. For all studies, samples were taken at indicated time points, serially diluted in PBS containing 0.01% tyloxapol and plated onto 7H11 OADC agar plates. Plates were incubated at 37 °C for 4 weeks to determine CFU counts.

### 2.10. ROI Assay

ROI production in *Mtb* strains (OD_600_ ~1.0) was measured using the dihydroethidium ROI sensing dye (DHE, excitation/emission at 500/605 nm). *Mtb* strains were cultured in Middlebrook 7H9 media supplemented with 0.2% glycerol and 0.01% tyloxapol at 37 °C with either 10% OAD (oleic acid, albumin, dextrose) with/without 0.25 mM cumene hydroperoxide (CHP) or 10% OADC (oleic acid, albumin, dextrose, and catalase) with or without 60 × MIC of CFZ for 16 h in 4 replicates. After treatment, *Mtb* cultures were washed by centrifugation (3000× *g*) and resuspended in 1× PBS (pH 7.4) containing 10 µM DHE, incubated further for 20 min at 37 °C followed by two washes with PBS to remove residual extracellular dye. The fluorescence of DHE-stained cells was acquired with a FACS Aria III cell sorter using the 500 nm laser excitation, and BP 610/20 nm for emission acquisition (PerCP-Cy^TM^5.5). The cells were acquired at a constant flow rate of setting 4, a threshold rate of approximately 1000–2000 events per second, and 100,000 total events were recorded per sample. For analysis, the bacterial population was identified according to the forward and side scattering property of the population (FSC versus SSC). To obtain single cell populations, bacterial aggregation was removed from the data analysis using doublet discrimination from the FCS-height versus FSC-area plots. The percentage of DHE+ cells and mean fluorescent intensity were calculated with FlowJo^TM^ v10.4.2 (Tree Star, Ashland, OR, USA).

### 2.11. Identification of Proteins by LC-MS/MS

The entire Bi_2_S_3_-stained protein band was excised from the gel, rinsed with water, and cut into approximately 1 mm × 4mm pieces using a sterile scalpel. The gel slices were then rinsed with 100 mM ammonium bicarbonate solution and collectively transferred into a sterile Eppendorf^®^ LoBind (Eppendorf, Germany) 1.5 mL microcentrifuge tube. Five hundred microliters of acetonitrile (ACN) was added, and the sample was incubated on ice for 10 min. The sample was then briefly centrifuged, the acetonitrile removed, and 100 µL of 10 mM dithiothreitol (DTT) solution was added to rehydrate the gel pieces and reduce the proteins. The sample was incubated in 10 mM DTT solution at 56 °C for 30 min, removed, cooled to room temperature, and then 500 µL of ACN was added and the sample was incubated on ice for 10 min. The sample was then centrifuged, and the supernatant removed, then 100 µL of 55 mM iodoacetamide solution was added and the sample was incubated at room temperature for 30 min in the dark to facilitate protein alkylation. Following alkylation, 500 µL of ACN was added and the sample was incubated on ice for 10 min. All solution was then removed, and 200 µL of trypsin (Promega, sequence grade) solution at a concentration of 13 ng/mL was added to the gel slices; the sample was mixed gently and incubated at 4 °C for 2 h to allow the gel slices to re-hydrate and for the slow diffusion of trypsin into the polyacrylamide gel matrix. The samples were incubated in the trypsin solution at 37 °C overnight (18–24 h) for optimum in-gel protein digestion. The resulting peptides were extracted by adding 400 µL of 5% formic acid/acetonitrile (1:2, *v*/*v*) solution to the sample followed by 15 min incubation at 37 °C on a shaking heating block set at 450 rpm. The sample was briefly centrifuged, the supernatant transferred to a sterile microcentrifuge tube and dried using a SpeedVac concentrator (Labconco, New York, NY, USA) set at 40 °C. The extracted, dried peptides were then reconstituted in 50 µL of 5% formic acid solution, transferred to a glass vial, and 1 µL of sample was injected for nano-LC-MS/MS analysis.

The peptide digests were analyzed using a Thermo Q Exactive Orbitrap mass spectrometer coupled to a Dionex^TM^ UltiMate^TM^ 3000 UPLC system. The tryptic peptides were maintained at 6 °C in the autosampler and were separated on a 15 cm nano-capillary column (ID 75 μM) packed in the laboratory with Supelco^®^ (Supelco Inc., New York, NY, USA) 3.5 μM C18 stationary phase. A 45 min gradient from 1% acetonitrile, 99% water/0.1% formic acid, to 50% acetonitrile/water, 0.1% formic acid, flow rate 300 nL/min, was used for the analysis. Peptide fragment mass spectra were acquired using a full MS, data dependent MS2 Top 10 method. The MS RAW files were processed using Thermo Scientific™ Proteome Discoverer™ (Thermo Fisher Scientific, USA) 2.2 software and SEQUEST™ (Thermo Fisher Scientific, USA) peak-finding search engine application to compare the mass spectra to the *Mtb* FASTA database to identify relevant proteins and peptides. The method was set to consider carbamidomethyl modifications and methionine oxidation. The protein candidates were then screened for pyridoxal phosphate binding domains and potential roles in sulfur metabolism. A targeted method was constructed using the 5 strongest peptide fragment ions for the most likely candidate, Rv3684/Cds1, and the samples re-analyzed using this method to confirm the presence of this protein.

### 2.12. Preparation of Mycobacterial Genomic DNA

Genomic DNA was isolated from *Mtb* H37Rv as follows: *Mtb* H37Rv was grown to late log phase (OD_600_ = 1.0) in 50 mL 7H9 liquid media. Cells were harvested (2000× *g*, 20 min), the supernatant was discarded, and 6 mL of a freshly prepared solution of 3 parts chloroform to 1 part methanol added. Tubes were then vortexed for 1 min. Tris-buffered phenol (6 mL) was then added and the tube vortexed for a further 30 s. Finally, 9 mL of 4 mM guanidine thiocyanate solution was added, and the tubes inverted several times. After centrifuging at 2000× *g* for 15 min, the upper phase was removed, and an equal volume of isopropanol was added to precipitate genomic DNA. The DNA was collected by centrifugation and washed with 70% ethanol before being air-dried and suspended in 100 µL Tris-EDTA, pH 7.5.

### 2.13. Construction of Δcds1 and Δrv1077 Mycobacterial Strains

The *rv1077* (*cbs*) and *rv3684 (cds1)* knockout *Mtb* strains were generated by homologous recombination using specialized phage transduction according to Badarov, et al. (2002) [35] (see Appendix A). The *cds1* allelic exchange substrate (AES) phasmid was a kind gift from Michelle Larsson (Albert Einstein College of Medicine). The AES contained *cds1* disrupted by the hygromycin resistance gene. Briefly, the AES phasmid was amplified in *E. coli* DH5α and purified using a DNA Plasmid Miniprep Kit (Thermo). *M. smegmatis* was then transduced with the AES, and a high titer phage lysate was prepared. *Mtb* H37Rv was grown to an OD_600_ of ~0.8 and washed twice with buffer (50 mM Tris-HCl, pH 7.6, 150 mM NaCl, 10 mM MgCl_2_, 2 mM CaCl_2_), then mixed with the high titer phage lysate in a 1:1 (CFU:plaque forming units) ratio, and incubated at 37 °C overnight. After centrifugation (16,000× *g*, 10 min, 4 °C) the pellet was resuspended in 0.2 mL 7H9 media and plated on hygromycin-containing 7H10 agar. After 3 weeks at 37 °C, 5 individual colonies were inoculated in 7H9 media supplemented with 50 µg/mL hygromycin. The genomic DNA of each colony was extracted and gene deletion confirmed using PCR with primers Rv3684CF, Rv3684CR (*rv3684*) or Rv1077CF, Rv1077CR (*rv1077*), and UUT (See Appendix A).

### 2.14. Mycobacterial Complementation

The *cds1* ORF was PCR amplified from genomic *Mtb* DNA using KOD Xtreme HotStart DNA polymerase (Roche, New York, NY, USA) according to the manufacturer’s protocol and primers (Rv3684F and Rv3684R); see Appendix A. The PCR product and pMV261 vector were digested with *Bam* HI and *Cla* I (Thermo Fisher Scientific, USA), isolated using agarose purification, and ligated using T4 DNA ligase (NEB, USA) to produce pMV261::*hsp_60_-cds1* (see Appendix A). A second complementation vector was constructed. The ORF of *rv3682*, *rv3683* and *cds1*, with an additional 500 bp upstream region, was PCR amplified using primers ponABCF and ponABCR (see Appendix A). The amplicon was digested using *Bam*HI and *Cla*I and ligated into the pMV261 vector. Complementation vectors expressing *cds1* under the control of either the *hsp_60_* or native promoter were electroporated (Gene Pulser Xcell, Bio-Rad, USA) into the *Mtb* Δ*cds1* strain and transformants selected on 7H10 agar plates containing hygromycin (50 µg/mL) and kanamycin (25 µg/mL). These vectors were similarly electroporated into *M. smegmatis*. Complemented strains were grown in 7H9 media containing 25 µg/mL kanamycin.

### 2.15. Purification of Recombinant Proteins

The *cds1* 1041 bp open reading frame was PCR amplified using *Mtb* genomic DNA and the primers Rv3684CEF and Rv3684CER (see Appendix A). The PCR product was digested with *Nde* I and *Bam* HI and then ligated into the pET15b expression vector previously digested with *Nde* I and *Bam* HI. These restriction sites are in the pET15b MCS downstream of a 6-His coding region, resulting in the addition of a 6-His tag to the N-terminus of the encoded protein. The ligated construct (i.e., pET15b-*cds1*) was then verified by sequencing. *E. coli* BL21 (DE3) cells were transformed with pET15b-*cds1* and grown until the OD_600_ reached 0.5–0.6. Protein expression was induced by the addition of 0.4 mM of IPTG followed by growth overnight at 18 °C. The cells were pelleted by centrifugation at 5000 rpm for 10 min, sonicated, and the lysate used for protein purification using nickel-affinity resin (Bio-Rad, USA) by gravity chromatography. pET28b-EhOASS expressing recombinant OASS containing an N-terminal 6×His tag was overexpressed and purified from *E. coli* as described previously [36].

### 2.16. Cds1 Enzyme Kinetics

The rate of H_2_S production by purified recombinant Cds1 was monitored via formation of Bi_2_S_3_, which has an absorbance maximum at 405 nm [32]. Briefly, purified Cds1 enzyme (1 µg of enzyme in 20 µL buffer) and 180 µL of BC solution were mixed in a well of a flat bottom, clear 96-well plate (Corning Inc., USA). Bi_2_S_3_ formation was monitored at room temperature over 30 min (readings taken once per minute) using a Biotek Synergy H1 hybrid plate reader. The absorbance at 405 nm was converted into product concentrations using the Beer–Lambert equation. The molar absorption coefficient for Bi_2_S_3_ was determined to be 3156.9 M^−1^ cm^−1^ using Na_2_S.9H_2_O (Sigma–Aldrich, USA) as a standard. Initial velocities were calculated and plotted against Cys concentrations. The K_m_ and V_max_ values were determined in GraphPad Prism (version 8.4.3) using the Michaelis–Menten equation. The k_cat_ values were calculated by dividing V_max_ by the nanomoles of enzyme used in the reaction.

### 2.17. Identification of Cds1 Enzymatic Products by LC-MS/MS

Purified Cds1 was added to three separate reaction tubes containing 50 µL of 20 mM L-Cys in PBS (20 mM Na_2_HPO_4_, 100 mM NaCl, pH 7.5, and 20 µM PLP), and the reaction was allowed to proceed for 15 min at room temperature. The reactions were stopped by heating to 80 °C for 5 min and the tubes centrifuged at 15,000 rpm for 5 min. Ten microliter aliquots of each reaction solution were taken from the supernatant and analyzed by LC-MS/MS. We then identified the enzymatic products of Cds1 using ultra-high liquid chromatography coupled to high-resolution/high-accuracy mass spectrometry based on the analyte’s exact mass and HPLC retention time compared to authentic metabolite reference standards. A reference standard mixture of important negatively charged metabolites, including pyruvate and the study samples, were analyzed on the Q-Exactive LCMS system, as described elsewhere, using high-resolution molecular ion scans. The RAW files were subjected to post-run analysis using Skyline software along with a template constructed to monitor the relevant analytes. The exact theoretical mass for ^12^C pyruvate standard is 87.008768 (87.0088) Da. The standard was present in the molecular ion scans with the same mass in the standard mixture when subjected to analysis. The retention time for the pyruvate in the standard mix on the Aminex column (Bio-Rad, USA) used was 11.0 min. In the samples, a peak was observed with the same retention time and exact mass as the pyruvate standard, and this was considered sufficient evidence for verification of identity. Each sample was assayed in triplicate.

### 2.18. In Vitro CFU Assay

Thioglycollate-elicited peritoneal macrophages were isolated from C57BL/6 mice [23] and were plated at 1.0 × 10^6^ cells per well in 6-well plates. Cells were infected with *Mtb* strains at an MOI of 0.2 and incubated at 37 °C for 2 h, followed by washing cells twice to remove non-internalized bacilli (day 0 post infection). At 0, 1, 2, and 4 days, cells were lysed with DPBS containing 0.05% SDS, and CFU were determined by plating serial dilutions of lysates on 7H11 agar plates supplemented with 10% OADC. Plates were incubated at 37 °C with 5% CO_2_ for 4 weeks to determine CFU counts.

### 2.19. LC-MS/MS Targeted Metabolomics Analyses

*Mtb* strains in replicate were inoculated from freshly grown culture in 7H9 media (containing 0.01% tyloxapol, 0.2% glycerol, and 10% OADC) with 4 mM cysteine at an OD_600_ of approximately 0.05. These strains were cultured (~30 mL each) in inkwell bottles to an OD_600_ of ~0.8 at 37 °C with shaking (150 rpm). Each culture was pelleted by centrifuging at 3500× *g* for 10 min. The culture supernatant was discarded, and cells were washed three times with 7H9 containing 0.01% tyloxapol. After the last wash, the pelleted cells were resuspended in 5 mL of 7H9 media containing 0.01% tyloxapol, 0.2% glycerol, 10% OAS (oleic acid, bovine albumin, and NaCl), 0.2% [U-^13^C]-Glucose (Sigma–Aldrich, USA) or 0.01% tyloxapol, 0.2% glycerol, 10% OADS (oleic acid, bovine albumin, dextrose, and NaCl), 100 µM [U-^13^C]-cysteine (Cambridge Isotope Laboratories Inc., USA), or 7H9 media containing 0.01% tyloxapol, 0.2% glycerol, 10% OAD, and 0.25 mM CHP. After resuspension of the pellet, the cultures were then incubated overnight at 37 °C with shaking (150 rpm) and harvested by centrifugation at 3500× *g* for 3 min. Cells pellets were immediately snap-frozen on dry-ice for approximately 5 min and then thawed and prepared for lysis using a MagNA Lyser (Roche, USA) at 7000 rpm for 1 min, followed by cooling on ice for 4 min, repeated 3–4 times. The lysis was performed using a 1.8 mL solution of methanol, acetonitrile, and water in the ratio of 2:2:1). The lysate was then centrifuged at 15,000–17,000× *g* for 10 min. The supernatant was collected and filtered through a 0.22 µm filter. The recovered lysate was then vacuum concentrated to dryness (Eppendorf Concentrator Plus, Eppendorf, Germany) at 30 °C for 12 h. The dried lysate pellets were then resuspended in 200 µL of purified water. 100 µL of this suspension for each replicate was submitted for LC-MS/MS targeted organic acid (metabolites) analysis and 100 µL (50 µL resuspension plus 50 µL acetonitrile) was submitted for LC-MS/MS targeted amino acid analysis.

LC-MS/MS was used for relative quantification of each organic acid and amino acid. The LC-MS/MS sample analysis was performed using a Thermo Scientific Dionex^TM^ UltiMate^TM^ 3000 UHPLC system coupled to a Thermo Scientific Q-Exactive Mass Spectrometer with a HESI source. A sample volume of 1 µL was injected onto a Waters Xbridge^®^ BEH HILIC HPLC column (2.5 µm particles, 2.1 × 100 mm), column oven set at 40 °C and chromatographic separation was performed using gradient elution at a flow rate of 200 µL/min and total run time of 26 min. Mobile phase A contained water with 0.1% formic acid and mobile phase B contained acetonitrile with 0.1% formic acid. Data were acquired using full-scan MS (without HCD fragmentation) in positive mode over the *m/z* range 50–750 Da at a 70,000 resolution. A QC sample was prepared using 21 different amino acids at a concentration of 500 ng/mL to monitor amino acid retention time consistency and MS sensitivity. Each sample was spiked with deuterated alanine (D4-Alanine, Sigma–Aldrich, USA) as an internal standard to monitor processing efficiency and data normalization. The data were processed, and peak areas were calculated using Skyline 3.7 (MacCoss Lab, University of Washington, Seattle, WA, USA).

### 2.20. Total ATP Quantitation

ATP quantitation was performed on cell lysates prepared for use in our metabolomics analysis (see Section 2.19). ATP levels of cell lysates were measured using a Roche ATP Bioluminescence kit CLS II according to the manufacturer’s instructions. ATP levels were normalized to the protein concentration of each cell lysate.

### 2.21. Statistics

Unless specified otherwise in Section 2, all experiments were performed on 3–8 biological replicates, and the data were expressed as the mean ± SD or the mean ± SEM. Statistical significance of the data was determined using GraphPad Prism 8.4.3 (GraphPad Software, Inc., USA). Specific statistical tests are noted in the figure legends and include the Student’s unpaired *t*-test (two-tailed) and one-way or two-way ANOVA.

## 3. Results

### 3.1. H_2_S Production by Mycobacteria

Since H_2_S can be spontaneously generated by media components [8] and some H_2_S detection methods are prone to artifacts (e.g., non-specific reaction of lead acetate strips with sulfides [12]), we carefully controlled the experimental conditions under which H_2_S levels were measured in solution or in a headspace using validated techniques. Three different methods were employed to measure H_2_S production in mycobacteria: the widely used lead acetate [Pb(Ac)_2_] method, the bismuth chloride (BiCl_3_; (BC)) method [32,33], and a highly sensitive amperometric microsensor (Unisense A/S, Denmark) [37].

We first used lead acetate strips to detect the presence of H_2_S in the headspace of mycobacterial cultures (Figure 1a). This method is based on the reaction of lead acetate with H_2_S to form lead sulfide (PbS), seen as a dark-colored precipitate on the strip [38]. We assessed the production of H_2_S in slow-growing laboratory strains (*Mtb* H37Rv and *Mtb* CDC1551 (CDC)), clinical strains (drug-sensitive (DS) and multidrug-resistant (MDR)), *M. bovis* BCG (BCG), *M. bovis* (*Mbov*) as well as the fast-growing mycobacterial strain *M. smegmatis* (*Msm*), inoculated to the same optical density. Notably, all mycobacteria tested produced H_2_S (Figure 1a). Since H_2_S production appears to be growth- and strain-dependent, we quantified lead sulfide formation on Pb(Ac)_2_ strips and normalized the values to culture density as described previously [31] (Figure 1b). BCG*, Mbov*, and the DS *Mtb* clinical strain produced significantly less H_2_S than the laboratory strains *Mtb* H37Rv and CDC. Notably, slow-growing pathogenic *Mtb* strains, particularly the MDR strains, produced the highest levels of H_2_S, whereas H_2_S production in *Msm* was virtually undetectable (Figure 1b).

To verify the results shown in Figure 1a,b, we employed a BC microplate assay to detect H_2_S in mycobacterial culture media and cell lysates. The BC microplate assay relies on the reaction of BiCl_3_ with H_2_S to generate bismuth sulfide (Bi_2_S_3_), a brownish black precipitate [32]. Importantly, BC-based H_2_S measurements are not influenced by cell proliferation, since the high concentrations of EDTA and Cys required for the assay inhibit growth. Furthermore, EDTA prevents the spontaneous generation of H_2_S from Cys and iron [8]. In the BC assay, we observed that only live *Mtb* produced H_2_S and not heat-killed bacilli or media alone (Figure 1c), and that H_2_S levels positively correlated with the concentrations of Cys present in the growth medium (Figure 1d; Appendix A). Several H_2_S-producing enzymes use Cys as a substrate and pyridoxal 5-phosphate (PLP) as a co-factor [18,19]. As shown in Figure 1e, addition of PLP significantly increased H_2_S production in *Mtb* lysates containing various concentrations of Cys. This suggests that the BC-based method is effective for measuring H_2_S in culture medium and cell lysates, and that at least one PLP-dependent enzyme is responsible for H_2_S production in *Mtb*.

Next, we attempted to characterize the H_2_S-producing enzymes in *Mtb* by using specific enzyme inhibitors. Aminooxyacetic acid (AOAA) is an inhibitor of PLP-dependent enzymes, including cystathionine β-synthase (CBS) and cystathionine γ-lyase, whereas DL-propargylglycine (PAG) inhibits only cystathionine γ-lyase (CSE) [39]. Addition of AOAA resulted in a concentration-dependent reduction of H_2_S in intact cells (Figure 1f) and in lysates (Figure 1g). In contrast, H_2_S levels were not significantly altered by PAG in either intact cells (Figure 1h) or lysates (Figure 1i). Again, these findings suggest that at least one PLP-dependent enzyme is a major contributor to H_2_S production in *Mtb*.

To further support these findings, we used a highly sensitive amperometric H_2_S microsensor (Unisense, A/S, Denmark) to directly detect H_2_S. Using this method, we observed that, compared to untreated controls, the addition of 1 mM Cys resulted in a three-fold increase in H_2_S levels in the media of dispersed *Mtb* cultures or in the corresponding cleared media containing pelleted cells after 72 h of incubation (Figure 1j). In addition, this microsensor allowed us to monitor H_2_S production in *Mtb* lysates using a single Cys concentration (Figure 1k) or increasing Cys concentrations in real time (Figure 1l), supporting our findings shown in Figure 1e. In contrast, when AOAA was pre-incubated with the lysate, no H_2_S was produced following addition of Cys (Figure 1m), indicating complete inhibition of H_2_S-producing activity.

In summary, using three different methods, we provide evidence that laboratory and clinical strains of *Mtb* produce H_2_S. We showed that this activity is PLP-dependent, is inhibited by AOAA and not PAG, and uses Cys as a sulfur source. An intriguing finding was the significant variation in H_2_S production among MDR and laboratory strains of *Mtb*, which is growth- and species-dependent. These findings have important implications for routine culturing of *Mtb* and improved diagnostics.

### 3.2. Identification of H_2_S-Producing Enzymes in Mtb

We searched the Kyoto encyclopedia of genes and genomes (KEGG) database [25] for enzymes encoded by *Mtb* H37Rv involved in H_2_S production. We identified 11 enzymes putatively involved in the metabolism of sulfur, sulfur-containing amino acids, and H_2_S (Appendix A). CBS is a well-studied enzyme that produces H_2_S in mammalian and bacterial cells, and *Mtb* encodes an ortholog of CBS, Rv1077 [25] (Appendix A). To determine the role of Rv1077 in H_2_S production, we deleted *rv1077*/*cbs* in *Mtb* H37Rv (Δ*cbs*) using specialized phage transduction [35]. Analysis of H_2_S production in intact Δ*cbs* cells or lysates revealed that H_2_S production in Δ*cbs* was not reduced compared to wild-type (WT) *Mtb* (Appendix A).

We then pursued a biochemical approach to identify H_2_S-producing enzymes in *Mtb*. In this approach, equal amounts of mycobacterial lysates were resolved on non-denaturing polyacrylamide gels and enzymatic production of H_2_S was detected by applying the BC assay solution directly to the gel [40]. Figure 2a shows the formation of Bi_2_S_3_ resulting from H_2_S production in various mycobacteria, seen as dark-colored bands of different intensities in the gel. Notably, addition of AOAA prior to the BC assay solution delayed the emergence and reduced the intensity of the major H_2_S-producing band (Figure 2b), consistent with the reduction in H_2_S production observed with AOAA in microplate assays (Figure 1f,g). Further, H_2_S production was markedly increased in the presence of PLP, suggesting that the prominent dark band contained the same enzyme assayed in Figure 1e, where PLP also increased the H_2_S-producing activity. Moreover, in the absence of PLP, the H_2_S-producing activity was completely abolished with AOAA (Figure 2b).

We next performed in-gel tryptic digestion of the main H_2_S-producing band from *Mtb* (Figure 2c) followed by LC*-*MS/MS analysis. Several overlapping peptide fragments were identified with high confidence, which identified Rv3684 as the putative H_2_S-producing enzyme (Figure 2c). Notably, Rv3684 was annotated as belonging to the Cys synthase/cystathionine β-synthase protein family (Mycobrowser.epfl.ch) (Appendix A). Importantly, we noticed an annotation ambiguity regarding the exact open reading frame (ORF) of rv3684. Based on our LC MS/MS data, we found that the actual start codon of the rv3684 ORF overlaps the stop codon of rv3683, and that these ORFs are in different coding frames (Appendix A). On this basis, we conclude that the rv3684 ORF has been incorrectly annotated in Mycobrowser.epfl.ch, resulting in the omission of the first 66 nucleotides.

CBS and CSE can use Cys as a substrate to generate H_2_S and other products such as serine (CBS) or pyruvate (CSE) (Figure 2d). To characterize the Rv3684 enzymatic activity, Rv3684 was overexpressed in *E. col**i* and purified (Figure 2e). Using Cys as a substrate, we determined that Rv3684 produces H_2_S and pyruvate, as shown by the in-gel BC assay (Appendix A) and LC-MS/MS analysis (Figure 2f), indicating this enzyme is functionally distinct from CBS. We next determined the initial velocities of H_2_S production at increasing Cys concentrations via the BC assay. From these data, we determined the K_m_ of Rv3684 to be 11.26 ± 0.75 mM with a k_cat_ of 78.71 ± 12.72 S^−1^ (Figure 2g). These data strongly suggest that Rv3684 catalyzes the conversion of Cys to form H_2_S, pyruvate, and ammonia in an α- and β-elimination reaction analogous to CSE (Figure 2h) [19]. Hence, we designated Rv3684 as a *c*ysteine *d*e*s*ulfhydrase enzyme (Cds1).

In sum, we employed a non-denaturing in-gel BC assay that demonstrates the activity of H_2_S-producing enzymes in mycobacterial lysates and identified Cds1 as an H_2_S-producing enzyme in *Mtb*. Identification of the enzymatic products indicates that Cds1 is a cysteine desulfhydrase that generates H_2_S and pyruvate from Cys.

### 3.3. Genetic Disruption of rv3684 Reduces H_2_S Production and Slows Mtb Growth in the Presence of Cys

To characterize the *rv3684 (cds1)* genetic locus, we used *Mtb* CDC1551 transposon mutants *Tn::rv3682* (*ponA2*) and *Tn::rv3683*, positioned upstream of *cds1* (Figure 3a). Notably, we observed that these mutants exhibited consecutively reduced H_2_S production from equal amounts of cell lysate (Figure 3b), confirming that *cds1* encodes an H_2_S-producing enzyme and that this locus is subject to a strong polar effect. Next, we created a *cds1* deletion mutant in *Mtb* H37Rv (Δ*cds1*) using specialized phage transduction [35] (Appendix A). We observed no Bi_2_S_3_ staining indicative of H_2_S production in Δ*cds1* lysates in the in-gel BC assay (Figure 3c), and saw significantly reduced H_2_S production in intact Δ*cds1* cells (Figure 3d) and lysates (Figure 3e). Genetic complementation of *cds1* in Δ*cds1* cells, *Mtb*Δ*cds1::hsp_60_-cds1* (*comp*), restored H_2_S production (Figure 3c–e). As shown in Figure 3d, the amount of H_2_S production in WT *Mtb* and *comp* cells were virtually identical. However, H_2_S production in *comp* lysates exceeded that of WT *Mtb* (Figure 3c,e), suggesting that Cys availability is limiting in intact *comp* cells, and that Cys import likely influences H_2_S production in *Mtb*. Of note, H_2_S production was reduced, but not eliminated in Δ*cds1* cells (Figure 3d), suggesting the presence of additional H_2_S-producing enzymes in *Mtb*. This is consistent with the appearance of a second band in our in-gel BC assay (Figure 2a). In contrast, we observed no H_2_S production in Δ*cds1* lysates (Figure 3e). This was not unexpected, since H_2_S-producing enzymes may require specific substrates, cofactors (e.g., NADPH, NADH, and heme) and may be influenced by environmental conditions (e.g., oxygen) [41]. H_2_S production was further confirmed in cellular lysates (Figure 3f) and whole cells (Figure 4g) of each strain using the Unisense amperometric microsensor.

Since exogenous and host-derived H_2_S supports *Mtb* bioenergetics and growth [22], we determined the consequence of *cds1* deletion on *Mtb* growth in medium containing Cys, a major source of sulfur and a substrate of Cds1 for H_2_S production. Compared to WT and *comp* cells, Δ*cds1* cells exhibited a significant growth defect in the presence of 4 mM Cys (Figure 3h). This finding is analogous to studies in *E. coli*, where disruption of D-cysteine desulfhydrase activity renders the bacteria susceptible to D-cysteine [42]. This suggests that Cds1 can mitigate toxic levels of Cys by converting excess Cys into H_2_S.

We investigated the role of *cds1* in *Mtb* survival during macrophage infection (Appendix A) and observed no significant difference in bacterial burden between WT *Mtb* and Δ*cds1* cells. This was not unexpected, as *Mtb* encodes multiple enzymes that may produce H_2_S (Appendix A), and conclusive demonstration of a role for H_2_S in pathogenesis may require genetic disruption of multiple H_2_S-producing genes and assessment in an in vivo model for TB. Since *Mtb* relies on lipid catabolism in vivo [43], we also investigated whether the growth of Δ*cds1* cells was influenced by utilization of fatty acids as a sole carbon source. Medium containing glycerol as the carbon source showed optimal growth for all *Mtb* strains, and no significant growth differences were observed between WT *Mtb* and Δ*cds1* cells in the presence of other fatty acids as the sole carbon source (Appendix A). In contrast, H_2_S production by Δ*cds1* cells was significantly reduced compared to WT or *comp* cells in all media containing fatty acids as the sole carbon sources (Appendix A). Notably, H_2_S production by Δ*cds1* cells was prolonged in media containing acetate, propionate or butyrate compared to glycerol and cholesterol-containing media.

Bioinformatic analyses showed that a homolog of Cds1 with >76% identity is present in all mycobacterial species (Figure 3i). Phylogenetic analysis revealed that orthologs are also present in bacteria other than *Mycobacterium* spp. (Figure 3i) (e.g., *Pseudosporangium, Streptomyces*, and *Rhodococcus* spp.). Further phylogenetic analyses of other sulfur metabolizing enzymes in mycobacteria demonstrate that sulfur metabolizing enzymes, such as CysK2, CysM, and CBS, show substantial similarities to Cds1 (Figure 3j). Of note was the similarity between Cds1, which converts Cys to H_2_S, and CysK1 which utilizes H_2_S as a substrate to form Cys [44].

In summary, using a series of biochemical and genetic approaches, we demonstrated that *Mtb* produces H_2_S primarily through Cds1. We functionally complemented *Δ**cds1* cells and show that genes upstream of *cds1* exert a strong polar effect on Cds1 levels, suggesting that *cds1* is in an operon with *rv3682* (*ponA2*) and *rv3683*. Lastly, we show that Cds1 is conserved in virtually all mycobacterial species with orthologs present in numerous other bacteria.

### 3.4. Conferring Increased H_2_S-Generating Activity to M. smegmatis

*Msm* has been widely used as a fast-growing surrogate for *Mtb* in numerous genetic and secretion studies [45]. Further, we observed that *Msm* produced very little H_2_S in our Pb(Ac)_2_ assay (Figure 1a,b). We reasoned that heterologous expression of *cds1* in *Msm* would allow the rapid study of how endogenous H_2_S affects mycobacterial physiology. To this end, we transformed *Msm* with plasmids that express *cds1* under control of the *hsp*_60_ promoter (*Msm–hsp_60_–*c*ds1*), or the *Mtb* wild-type (“native”; *wt_p_*) promoter (*Msm–wt_p_–*c*ds1*) (see Appendix A). *Msm–hsp_60_–*c*ds1* and *Msm–wt_p_–*c*ds1* lysates generated Bi_2_S_3_ bands with a migration pattern similar to that in *Mtb* lysates. However, Bi_2_S_3_ formation substantially increased compared to WT *Mtb*, likely due to the fact of *cds1* overexpression from multi-copy expression plasmids (Figure 4a). We also observed a Bi_2_S_3_ band in WT *Msm* lysates, indicating the presence of an endogenously expressed H_2_S-producing enzyme with migration properties distinct from *Mtb* Cds1. This *Msm* H_2_S-producing band was not observed in *Msm–hsp_60_–*c*ds1* or *Msm–wt_p_–*c*ds1* lysates, possibly due to the altered transcriptional or post-transcriptional regulation in the presence of overexpressed Cds1. H_2_S production in *Msm–wt_p_–*c*ds1* and the *Msm–hsp_60_–*c*ds1* lysates using the BC assay (Figure 4b) or the Unisense microsensor (Figure 4c) was more rapid and robust than in *Msm* containing the empty vector (*Msm-*pMV261), suggesting overproduction of Cds1 as observed in Figure 4a. 

However, H_2_S production in intact Cds1-producing *Msm* cells measured over time (Figure 4d) or at an endpoint (Figure 4e) was markedly lower than in the corresponding lysates (Figure 4b,c), suggesting that the lower H_2_S levels observed in *Msm* cells may be due to the limitations in Cys transport.

We next examined the impact of overexpressed *cds1* on the growth of *Msm* cultured in media containing Cys (0, 8, and 16 mM). *Msm–hsp_60_–cds1* and *Msm–wt_p_–cds1* maintained a significant growth advantage over control *Msm-*pMV261 after 48 h in medium containing 8 mM Cys (Figure 4f,g). These results suggest that Cds1 confers resistance to Cys-induced toxicity in *Msm* by converting excess Cys into H_2_S. Hence, our data suggest that due to its low capacity for H_2_S production, *Msm* can be used as an effective surrogate for studying the effect of H_2_S on mycobacterial physiology.

### 3.5. Endogenous H_2_S Stimulates Respiration in Mtb

Several studies have shown that H_2_S can inhibit or stimulate mammalian respiration in a concentration-dependent manner [20,21]. Similarly, a recent study has shown that exogenous H_2_S can stimulate *Mtb* cellular respiration [22]. On the other hand, biochemical studies have shown that H_2_S inhibits purified *E. coli* cytochrome *bc_1_/aa_3_* [22,46] and that cytochrome *bd* oxidase is resistant to inhibition by H_2_S [36,47]. Hence, we hypothesized that endogenous H_2_S produced by Cds1 can modulate *Mtb* respiration. To test this hypothesis, we used extracellular metabolic flux analysis, a methodology developed for eukaryotic cells which we optimized for real-time, quantitative study of *Mtb* respiration [22,34].

First, we measured the basal oxygen consumption rate (OCR) of Δ*cds1*, which was ~40% lower than the basal rate in WT or *comp* (Figure 5a). Of note, a reduction in respiration of this magnitude is highly significant in bioenergetic terms and suggests that endogenous H_2_S stimulates *Mtb* respiration. Next, we sought to determine whether Cds1-generated H_2_S acts as an effector to directly modulate basal respiration in *Mtb*. Several lines of evidence indicate that this is the case. Firstly, addition of L-Cys leads to increased respiration in *Mtb* in a concentration-dependent manner (Figure 5b), whereas the OCR of Δ*cds1* cells is unchanged by the addition of 4 mM Cys, in contrast to WT and *comp* (Figure 5c). Secondly, deletion of *cbs* (*rv1077*) in *Mtb* (Δ*cbs*) did not significantly reduce the OCR compared to WT cells in the presence of 4 mM Cys (Appendix A). Thirdly, addition of AOAA, an inhibitor of PLP-dependent enzymes, such as Cds1, eliminated the Cys-stimulated increase in OCR in WT *Mtb* (Figure 5d). Likewise, addition of AOAA followed by Cys decreased respiration in *Mtb* cells (Appendix A), which was attributed to the inhibition of Cds1 and other PLP-dependent enzymes by AOAA. Importantly, AOAA alone has no significant effect on respiration in the absence of Cys (Appendix A). Taken together, these data suggest that Cds1-generated H_2_S is important for maintaining basal respiration in *Mtb*.

To further confirm that H_2_S is the endogenous effector molecule modulating *Mtb* respiration, we exposed Δ*cds1* cells to exogenous H_2_S and observed reversal of the respiratory defect in these cells (Appendix A). Next, we used an H_2_S-degrading enzyme, *O*-acetylserine sulfhydrylase [44] (OASS), in our bioenergetic assays (Appendix A). Using H_2_S and *O*-acetylserine (OAS) as substrates, OASS catalyzes a β-replacement reaction to produce Cys and acetate. Enzymatic activity of OASS was confirmed by monitoring a reduction in H_2_S concentration using the Unisense microsensor (Figure 5e). Sequential addition of NaHS increased H_2_S levels, whereas injection of purified OASS caused a rapid decrease in H_2_S levels after addition of the substrate, OAS (Figure 5e). On this basis, we posited that addition of OASS and its substrate OAS would deplete H_2_S produced by *Mtb*, resulting in a sustained or reduced OCR. Indeed, as shown in Figure 5f, our data suggest that addition of OAS and purified OASS to wells containing *Mtb* prevents the Cys-mediated increase in OCR via degradation of *Mtb*-generated H_2_S by OASS activity. Although a product of the OASS reaction is Cys, which can re-enter the cell as a substrate for H_2_S production, Cys transport across the membrane is rate-limiting. Hence, exogenous OASS depletes H_2_S faster than Cys can be transported into the cytoplasm and used for H_2_S production. Overall, OASS and OAS significantly reduced H_2_S in our experimental system. Lastly, cytoplasmic H_2_S may also be consumed by *Mtb* CysK1 using exogenous OAS as a substrate [44]. Indeed, addition of OAS and Cys to *Mtb* cells significantly reduced OCR compared to *Mtb* cells treated with Cys alone, suggesting CysK1 can consume excess H_2_S depending on intracellular levels of OAS (Figure 5g) and led to the model in Figure 5h.

We next examined the mechanism whereby H_2_S stimulates *Mtb* respiration. In a previous study [22], our findings in *M. smegmatis* and *Mtb* implicated cytochrome *bc_1_/aa_3_* and cytochrome *bd* (CytBD) in enhancing respiration when exposed to H_2_S, primarily because the *Mtb* electron transport chain (ETC) can rapidly reroute electrons to either of these oxidases (Figure 5i). To investigate whether Cys-generated H_2_S regulates respiration through cytochrome *bc_1_/aa_3_* and/or CytBD, we examined Δ*cydAB* cells (genetic knockout of *Mtb cytBD*) in the presence of Cys and the cytochrome *bc_1_/aa_3_* inhibitor Q203. Hence, the Δ*cydAB* strain of *Mtb* produces only a functional cytochrome *bc_1_/aa_3_* [48], and cellular respiration can be fully inhibited by Q203 [34]. Intriguingly, both WT *Mtb* (Figure 5j) and Δ*cydAB* cells (Figure 5k) show increased respiration when exposed to Cys, suggesting that H_2_S stimulates *Mtb* respiration via cytochrome *bc_1_/aa_3_* oxidase. However, *Mtb* cells treated with the cytochrome *bc_1_/aa_3_* oxidase inhibitor, Q203, and Cys showed a more profound increase in respiration (OCR% ~330; Figure 5l) compared to Δ*cydAB* cells (OCR% ~145; Figure 5k) at 1 mM Cys. This suggests that Cys-generated H_2_S stimulates *Mtb* respiration more strongly through cytochrome *bd* compared to cytochrome *bc_1_/aa_3_* oxidase. As expected, respiration was dramatically reduced when Δ*cydAB* cells were exposed to Q203 alone or Q203 and Cys (Figure 5m). While recognizing that the precise mechanism whereby H_2_S stimulates *Mtb* respiration is complex, these data provide strong evidence that Cds1 uses Cys as a source of sulfur to produce H_2_S, which modulates *Mtb* respiration predominantly via CytBD.

### 3.6. Mtb H_2_S Regulates Sulfur Metabolism

Our data demonstrate that Cds1 is a cysteine desulfhydrase that generates H_2_S from Cys (Figure 2f). Hence, Cds1 may serve to eliminate toxic levels of Cys, which can induce oxidative stress via the Fenton reaction [9,49]. Not surprisingly, Cys can be rapidly oxidized to cystine (the disulfide form of Cys; Cys_ox_) [50]. Overall, the enzymatic properties of Cds1 are consistent with our data showing that Δ*cds1* cells exhibit reduced growth in medium containing toxic concentrations of Cys (Figure 3f). To test the hypothesis that Cds1-generated H_2_S regulates cellular sulfur metabolism, we performed carbon tracing experiments using [U-^13^C]-Cys (100 µM in 7H9 medium; capable of inducing H_2_S production [Appendix A]), to track the fate and incorporation of Cys carbons into metabolites. We subsequently examined metabolite abundance and the carbon isotopologue distribution (CID) of targeted sulfur metabolites (Figure 6a–g) in WT *Mtb*, Δ*cds1*, and *comp* cells.

We observed an overall increase in the abundance of sulfur pathway metabolites cystathionine (Cth), O-succinyl homoserine (OSH), homoserine (Hse), homocysteine (Hcy), Met, Cys, and Cys_ox_ in Δ*cds1* cells (Figure 6a–g). A likely explanation for the build-up of these sulfur metabolites is that sulfur atoms cannot be dissipated via Cds1-mediated release of H_2_S, thereby generating back pressure and a subsequent metabolite build up. Exogenous [U-^13^C]-Cys rapidly converted into Cys_ox_, which accumulated significantly more in Δ*cds1* cells (Figure 6a) compared to the controls due to the lack of Cds1 activity and a subsequently impaired capacity to recycle sulfur atoms via CysK1 and O-acetyl serine (OAS) back to Cys (Figure 6b). In Δ*cds1* cells, most of the carbons in Cys were unlabeled (Figure 6b), which suggests that most carbons for de novo synthesis of Cys originated from glucose or glycerol, leading to a small but significant increase in unlabeled Cys. Lastly, the significant changes in CID of M + 5 and M + 6 species in the OSH metabolite pool and increased abundance of Cth (Figure 6c), OSH (Figure 6d), and Hse (Figure 6e) are indicative of reduced flux of [U-^13^C]-Cys carbons in the sulfur pathway leading to the subsequent buildup in Δ*cds1* cells. Notably, Hcy was almost completely labeled, pointing to the importance of this metabolite in the Cys–Cth conversion step and anabolism of Hcy via MetC (Figure 6g).

To complement our [U-^13^C]-Cys results, in an independent experiment, we used [U-^13^C]-glucose as the carbon tracer. The rationale for using labeled glucose is twofold; firstly, unlike [U-^13^C]-Cys, this approach allowed us to trace the incorporation of glucose-derived carbons into sulfur metabolites. Secondly, it allowed us to trace the incorporation of glucose carbons into de novo-synthesized Cys, which is a substrate for H_2_S production in the absence of an exogenous substrate. Importantly, similar to our [U-^13^C]-Cys results (Figure 6a–g), we also observed an accumulation of sulfur pathway metabolites (e.g., Cys, Cth, OSH, Hse, and Met) in Δ*cds1* cells compared to WT and *comp* (Figure 6h–m). Hence, the [U-^13^C]-glucose tracing data complement our Cys carbon tracing data (Figure 6a–g), as they show that de novo synthesis of Cys is required for H_2_S production, which is not possible to demonstrate using [U-^13^C]-Cys, since it is a substrate for H_2_S production (Appendix A).

Intriguingly, all Cys carbons were unlabeled, and Cys showed increased abundance in Δ*cds1* cells (Figure 6i), suggesting that >50% of Cys_ox_ carbons originated from glucose, whereas most of the Cys carbons originated from glycerol when cultured in 7H9 medium containing [U-^13^C]-glucose and glycerol. Increased accumulation of Hse was noted in Δ*cds1* cells despite reduced flux, which was evident by an increase in CID of M + 0 species and decreases in CID of M + 1, M + 2, and M + 3 species (Figure 6l).

In summary, these data demonstrate that Cds1 is important for maintaining homeostatic levels of sulfur pathway metabolites through the production of H_2_S, which allows recycling of sulfur atoms back to Cys and metabolites in the sulfur pathway. Lack of Cds1 activity triggers metabolic dysregulation of key sulfur pathway metabolites as is evident by the corresponding build-up of Cys, Cys_ox_, Cth, OSH, Hse, Hcy, and Met. Hence, our data suggest that Cds1-generated H_2_S functions as a sink to maintain sulfur homeostasis.

### 3.7. Mtb H_2_S Regulates Central Metabolism

There is a surprising lack of targeted metabolomic studies of sulfur metabolism in bacteria, as we could identify only two such studies [51,52]. Both studies reported that different sulfur fuel sources affect glycolysis, the TCA cycle, amino acid levels, and redox couples such as glutathione and MSH. Since we have shown that Cds1/H_2_S modulates growth (Figure 3f) and respiration (Figure 5c,g), which is tightly linked to metabolism, we tested the hypothesis that Cds1-generated H_2_S modulates *Mtb* central metabolism. To test this hypothesis, we cultured WT *Mtb*, *Δcds1*, and *comp* cells in 7H9 medium containing [U-^13^C]-Cys and examined metabolites in glycolysis, the pentose phosphate pathway (PPP), TCA cycle, and all amino acids (Figure 7). A striking observation was the significant increase in most amino acids (Figure 7a), glycolytic metabolites, PPP metabolites (6-phosphogluconate (6PG) and ribulose-5-phosphate (R5P)), and TCA intermediates in Δ*cds1* cells compared to WT and *comp* cells (Figure 7b), which strongly suggests that H_2_S plays a key role in modulating central metabolism. Importantly, we observed a similar, and significant increase in amino acids in Δ*cds1* cells compared to WT and *comp* cells using [U-^13^C]-glucose as the carbon tracer (Appendix A). This was not entirely unanticipated as endogenous (Figure 5g) and exogenous H_2_S regulate respiration (OXPHOS) in *Mtb* [22]. Also, H_2_S directly targets enzymes in the glycolytic pathway through persulfidation to modulate their activity [53,54] and was shown to suppress glycolysis in *Mtb*-infected macrophages [23]. The overall reduced labeling of amino acids and glycolysis/TCA cycle metabolites in Δ*cds1* cells is consistent with the concept that these metabolites obtain their carbons mainly from glucose or glycerol and not Cys. We observed a similar accumulation of sulfur pathway metabolites. On the other hand, F16BP and PEP were substantially labeled (~50%) in all three strains, which are the product and substrate of the first and last rate-limiting steps in glycolysis, respectively. This points to phosphofructokinase-1 (Pfk1) and pyruvate kinase (PykA) as important rate-limiting flux control points for the metabolism of Cys carbons into central metabolism. Further, to ascertain whether increased glycolytic and TCA metabolites translate into changes in ATP levels, we quantified ATP in cells with or without Cys (100 μM). As shown in Figure 7c, ATP levels were significantly increased in Δ*cds1* cells cultured with Cys, whereas no differences in ATP levels were observed in the absence of Cys.

In summary, our targeted metabolomics data demonstrate that Cds1-generated H_2_S suppresses the central metabolism, which is evident by increased levels of glycolytic and TCA cycle metabolites and amino acids in Δ*cds1* cells. In addition, the reduction in H_2_S levels in Δ*cds1* cells, which is associated with decreased respiration (OXPHOS) (Figure 5a), triggers a compensatory glycolytic response to maintain bioenergetic homeostasis as was evident by increased ATP levels in Δ*cds1* cells. Collectively, our data demonstrate that endogenous H_2_S functions as a modulator of the balance between OXPHOS and glycolysis in *Mtb*.

### 3.8. Endogenous H_2_S Exacerbates Oxidative Stress and Regulates Intracellular Redox Homeostasis

Several studies have reported a role for H_2_S in the modulation of redox homeostasis [55,56,57]. In *E. coli*, endogenously produced H_2_S maintains redox homeostasis by rendering *E. coli* resistant to oxidative stress [14]. Hence, we tested the hypothesis that Cds1-generated H_2_S modulates redox homeostasis in *Mtb* by measuring the abundance and flux of the two major redox couples EGT and MSH via metabolomics and by quantifying reactive oxygen intermediates (ROI) by flow cytometry [22]. Since catalase is present in standard 7H9 growth medium to protect cells against toxic peroxides and promote growth, we examined the levels of ROI in *Mtb* cells (Appendix A) in the presence and absence of catalase. In medium containing catalase, we observed ~43% more ROI-positive Δ*cds1* bacilli compared to WT and *comp* (Figure 8a), suggesting that in WT *Mtb*, H_2_S may function as an antioxidant as has been reported in mammalian cells [18]. In the absence of catalase, the percentage of ROI-positive cells increased by 4–10-fold in all strains compared to cells in the presence of catalase (Figure 8a). Of note, ~25% fewer Δ*cds1* bacilli were ROI-positive compared to WT or *comp* cells, suggesting that H_2_S functions as a pro-oxidant under these conditions.

Next, since Cys is a sulfur-containing precursor of MSH and EGT, we examined the abundance and CID of these redox couples in Δ*cds1* cells cultured in 7H9 medium containing [U-^13^C]-glucose (Figure 8b). Intriguingly, EGT and MSH levels were reduced in Δ*cds1* cells. Further, the increase in the CID of M + 1, M + 2, M + 3, M + 4, and M + 1 and M + 2 species in EGT and MSH, respectively, suggests increased carbon scrambling and reduced flux of carbons in response to the failure to recycle H_2_S back to Cys in Δ*cds1* cells. This agrees with the lack of increase in fully labeled species of EGT (M + 9) and MSH (M + 17). Therefore, our data demonstrate that H_2_S is necessary for maintaining homeostatic levels of MSH and EGT and, thus, redox balance. To further examine the role of H_2_S in *Mtb* redox homeostasis, we exposed *Mtb* to the oxidant cumene hydroperoxide (CHP) without catalase and monitored ROI production and cell viability. After 16 h of exposure to 0.25 mM CHP, Δ*cds1* cells had significantly fewer ROI-positive cells (Figure 8c) with increased survival (Figure 8d) compared to WT and *comp* cells. Notably, we observed no significant differences in the abundance of MSH or EGT between WT, *comp*, and Δ*cds1* cells after exposure to CHP (Appendix A).

Collectively, these data show that H_2_S can function either as a pro-oxidant or antioxidant depending on the experimental conditions. Further, the survival data indicate that reduced homeostatic levels of H_2_S promote survival during oxidative stress in the absence of catalase. Hence, under oxidative stress conditions, endogenously produced H_2_S in *Mtb* functions as a pro-oxidant. Lastly, our data demonstrate that Cds1-generated H_2_S modulates the levels of the major redox couples, EGT and MSH.

### 3.9. Endogenous H_2_S Increases Mtb Susceptibility to Clofazimine (CFZ) and Rifampicin (RIF)

H_2_S has been shown to alter antibiotic susceptibility in several bacterial pathogens including *E. coli, Bacillus anthracis, Pseudomonas aeruginosa,* and *Staphylococcus aureus* [14,15]. Hence, understanding the impact of endogenous H_2_S on *Mtb* drug susceptibility may have clinical implications. We posited that Cds1-generated H_2_S modulates susceptibility to the anti-TB drug clofazimine (CFZ), a known ROI generator [34]. After 24 h of CFZ exposure at 60× MIC, ~30% fewer Δ*cds1* cells were ROI-positive compared to WT or *comp* cells (Figure 8e). Further, we observed significantly increased CFU-based survival of Δ*cds1* versus WT control cells after 8 days of CFZ exposure (Figure 8f). These differences in survival between Δ*cds1* and WT and *comp* strains exposed to CFZ were significant, albeit modest, which may be due to the fact that H_2_S production was reduced but not eliminated in Δ*cds1* cells (Figure 3d). Hence, genetic knockout of multiple genes involved in H_2_S biosynthesis may further decrease *Mtb* sensitivity to CFZ. Regardless, these CFZ-exposure data indicate that endogenous H_2_S increases ROI in WT *Mtb* that contributes to CFZ susceptibility.

Exogenous H_2_S has been shown to modulate antibiotic susceptibility in a range of bacterial pathogens [14,15,18,55]. To determine whether exogenous H_2_S plays a role in *Mtb* drug susceptibility, CFZ-treated *Mtb* cells were exposed to H_2_S via addition of NaHS. Exogenous H_2_S significantly reduced survival of CFZ-treated *Mtb* cells (Figure 8g,h). Likewise, *Mtb* cells treated with rifampicin (RIF) or isoniazid (INH) for five days were exposed to exogenous H_2_S. As shown in (Figure 8i), H_2_S significantly increased susceptibility to RIF but not to INH. In summary, these data suggest that H_2_S increases *Mtb* susceptibility to CFZ and RIF through its function as a pro-oxidant.

## 4. Discussion

This study elucidated several previously unrecognized physiological features of *Mtb*. Using three different methods, we showed that laboratory and MDR and DS clinical *Mtb* strains produced H_2_S as do non-pathogenic slow- and fast-growing mycobacterial strains. We then identified the genetic locus *cds1* (*rv3684*) and confirmed that the gene product, Cds1, is a PLP-dependent H_2_S-producing enzyme. Importantly, we demonstrated that endogenous H_2_S can influence *Mtb* bioenergetics by enhancing respiration, primarily via CytBD, and by modulating the balance between respiration (OXPHOS) and glycolysis. Further, we described a plausible mechanism by which *Mtb* mitigates oxidative stress by converting excess Cys into H_2_S, which is released and then recycled. Our findings point to a paradigm whereby *Mtb*-generated H_2_S, together with host-generated H_2_S, exacerbates TB disease by dysregulating host immunity [23]. Lastly, since H_2_S production by *Mtb* has been an overlooked confounder in routine culturing of *Mtb*, we anticipate our findings to have a broad practical impact in the TB field.

Since the discovery of *Mtb* by Robert Koch in 1882, formal proof that *Mtb* produces H_2_S has been lacking. One likely reason is that since *Mtb* is highly contagious and spread through aerosols, smelling of cultures is a significant health risk and extreme measures are taken to avoid inhalation. Nonetheless, several biochemical studies [26,27,28,29,30] as well as homologues of H_2_S-producing enzymes in the *Mtb* genome have provided strong circumstantial evidence that *Mtb* is a likely producer of H_2_S. Furthermore, complicating factors are that H_2_S is a difficult molecule to study because measuring it is complex [6] and H_2_S can be spontaneously generated from media components [8]. Hence, we employed multiple approaches to provide compelling evidence that *Mtb* produces H_2_S.

Our findings that two clinical MDR *Mtb* strains produced the highest levels of H_2_S, followed by laboratory strains and clinical DS strains, have considerable clinical importance for several reasons. Firstly, many clinical MDR and XDR *Mtb* strains are notoriously difficult to culture due to extremely slow growth. Since high concentrations of H_2_S inhibit respiration [20,58], it is not unreasonable to propose that excessive H_2_S production contributes to the slow growth of many of these strains, even when cultured in the absence of Cys. Secondly, our findings that endogenous levels of H_2_S suppress central metabolism suggest that excessively high levels of H_2_S produced by MDR *Mtb* strains could reprogram metabolism by shifting the balance between OXPHOS and glycolysis, ultimately leading to an energetically impaired state that inhibits growth. Thirdly, the large variation in H_2_S production among clinical strains likely reflects functional differences that are due to the fact of SNPs or genomic rearrangements that contribute to strain-specific transcriptional regulation. Lastly, since H_2_S is widely used as a diagnostic test for bacteria, there is potential for the development of H_2_S-based diagnostics for *Mtb*, e.g., detection of H_2_S in the exhaled breath or sputum of TB patients.

How do these findings contribute to a more accurate understanding of *Mtb* physiology and pathogenesis? In two recent studies, it was shown that host-derived H_2_S exacerbates *Mtb* disease in the mouse model of TB [22,23] and that CSE and 3MST protein levels are markedly increased in human TB lesions that surround necrotic granulomas and cavities [23]. Hence, it was proposed that because of the strong immunomodulatory activity of H_2_S, excessive host H_2_S production triggered by *Mtb* infection dysregulates immunity to promote disease [23]. Considering the findings in this study, we posit that *Mtb*-generated H_2_S may act as a signaling molecule in the host, further contributing to excessive H_2_S levels in vivo to exacerbate disease. This is possible since Cys concentrations range from ~128–250 μM in human cells and plasma [59,60]. Although a role for *Mtb cds1* in TB disease has not yet been reported, our genetic knockout (Figure 3d) and other data (Appendix A) indicate that more than one enzyme contributes to H_2_S production in *Mtb*. Therefore, multiple gene knockouts may be necessary to convincingly demonstrate a role for endogenously-produced H_2_S in TB disease.

Discovery of a gene responsible for H_2_S production in *Mtb* cells establishes a paradigm for how H_2_S modulates *Mtb* physiology. While homologues of CBS and CSE are encoded in the *Mtb* genome, these proteins were not identified in our non-denaturing in-gel BC assays. However, this could be explained by the fact that the conditions of our in-gel BC assay were suboptimal for certain enzymes, and that alterations in environmental factors (substrate, pH, temperature, oxygen, etc.) or redox-dependent cofactors (e.g., NADH, NADPH, and heme) would allow for the detection of additional enzymes.

Based on our cell-based assays, Cds1 contributes significantly to H_2_S production and showed increased activity when cells were exposed to Cys. Since the total sulfur atom concentration in bacteria (e.g., *E. coli*) is ~130 mM and intracellular Cys serves as the primary supplier of sulfur atoms [47], it is possible that Cds1 shuttles sulfur atoms among metabolites to meet cellular demand and to maintain redox balance. Indeed, Cds1 protects *Mtb* and *Msm* against toxic concentrations of Cys, providing insight into the mechanisms, whereby Cds1 mitigates intracellular oxidative stress triggered by Cys, which feeds the Fenton reaction to generate free radicals [9]. Therefore, our data (Figure 3f, Figure 6 and Figure 7) support the concept that the Cds1-mediated conversion of Cys into H_2_S functions as a sink for excess Cys. This concept is further supported by studies showing that high levels of Cys inhibit growth [61], induce ROI production, and lead to DNA damage in *E. coli* [9] and *Mtb* [49].

Like other small nonelectrolytes, such as •NO, CO, and O_2_, at a gas/liquid interface H_2_S in solution (as a dissolved solute) readily “escapes” solvation and volatilizes. Likewise, as a gas at this interface, it will readily dissolve and, so, with a confined headspace, a dynamic equilibrium of volatilization/dissolution will exist, defined quantitatively by Henry’s Law. As pointed out elsewhere [18], the ability of H_2_S to exist as a gas has functional biological relevance only in the presence of such a gas/liquid interface, which is also true for •NO, CO, and O_2_ [62]. As demonstrated previously [63], the contribution of volatilization in vitro will be determined by the specific experimental configuration. In the lung, where such an interface is central to organ function, the “headspace” is not confined. Therefore, according to Le Chatelier’s principle, the process of ventilation will serve to pull this equilibrium in the direction of volatilization and, therefore, may well serve to amplify the importance of desulfhydration as a sink for H_2_S (and its conjugate bases HS^−^ and S^2−^) in vivo. The relative importance of volatilization as a sink will depend on its magnitude compared to other competing mechanisms of consumption in the organism and tissue [64].

Our finding that Cds1 generates H_2_S and pyruvate using Cys as a substrate suggests that Cds1 is catalytically similar to mammalian and bacterial CSE, which converts Cys into H_2_S, pyruvate, and ammonia [19]. Elegant biochemical studies have shown that *Mtb* Rv1079 possesses both CSE and cystathionine γ-synthase (CGS) activity [27]. However, Rv1079 lacks Cys desulfhydrase activity, and these authors speculated that an as yet uncharacterized enzyme performed this important role [27]. We provide genetic and biochemical evidence that Cds1 fulfils this function through its Cys desulfhydrase activity, which adds to our understanding of how *Mtb* detoxifies Cys.

An unusual feature of H_2_S is its ability to stimulate bioenergetics at low concentrations and inhibit respiration at higher concentrations. In a recent study, we demonstrated that low concentrations of exogenous H_2_S stimulate *Mtb* respiration and growth [22]. In mammals, H_2_S has been implicated in reversibly inhibiting cytochrome c oxidase (Complex IV) at high concentrations, and, conversely, stimulating mitochondrial respiration at low concentrations [20,21,65,66]. Our finding that Cds1-generated H_2_S is an effector molecule that modulates basal respiration reveals a previously unknown facet of *Mtb* physiology. This is supported by genetic and bioenergetic data showing a ~40% reduction in the basal respiration of Δ*cds1* cells compared to WT *Mtb*.

By exploiting a *Mtb cytBD* mutant and pharmacological inhibition of cytochrome *bc_1_/aa_3_*, our findings provide insight into how H_2_S can stimulate respiration. Contrary to studies in *E. coli* that show cytochrome *bo_3_* oxidase is susceptible to H_2_S and its two *bd* oxidases are resistant to H_2_S [36,47], our current data suggest that Cys-generated H_2_S stimulates respiration via both *Mtb* cytochrome *bc_1_/aa_3_* and CytBD, consistent with our previous findings [22]. For example, in the presence of Cys, respiration is dramatically increased when cytochrome *bc_1_/aa_3_* oxidase is inhibited with Q203 compared to respiration in the *Mtb cytBD* mutant. Hence, CytBD plays a more prominent role than cytochrome *bc_1_/aa_3_* in H_2_S-mediated stimulation. Determining whether H_2_S directly binds to one or both oxidases in *Mtb* to stimulate respiration will require further investigation. *Mtb* and *E. coli* respond differently to H_2_S, likely because they occupy vastly different niches. *Mtb* is an obligate aerobe that colonizes the human lung, whereas *E. coli* is a facultative anaerobe and ubiquitous member of the human gut microbiota that is exposed to millimolar concentrations of H_2_S. More importantly, unlike other bacteria, *Mtb* has the unusual ability to rapidly reroute ETC flux to either cytochrome *bc_1_/aa_3_* or CytBD [34] depending on which oxidase is inhibited. This remarkable plasticity allows *Mtb* to effectively respond to host gases, such as NO and CO, which inhibit respiration, and H_2_S, which can stimulate respiration.

Respiration is directly linked to central metabolism, and not surprisingly, Cds1-generated H_2_S suppresses glycolysis in WT *Mtb* as was evident by increased levels of numerous glycolytic metabolites in Δ*cds1* cells. We propose two plausible mechanisms whereby H_2_S could modulate the balance between OXPHOS and glycolysis (Figure 8j): Firstly, reduced respiration (OXPHOS) in Δ*cds1* cells triggers a compensatory glycolytic response, i.e., substrate level of phosphorylation to meet the bioenergetic demands for ATP. This is not an unusual metabolic response, as it was previously posited that bedaquiline-mediated inhibition of OXPHOS leads to a compensatory induction of glycolysis to meet the demand for ATP through substrate-level phosphorylation [34,67]. In addition, recent studies on *Bacillus* and *Staphylococcus* spp. showed that glycolysis can reverse polymyxin B-mediated ATP depletion that resulted from dysregulation of OXPHOS [68,69]. Secondly, H_2_S can directly target enzymes in the glycolytic pathway through persulfidation to modulate their activity [53,54]. This is supported by recent studies showing that H_2_S suppresses glycolysis in macrophages upon *Mtb* infection [23].

Respiration and central metabolism are also linked to redox balance (via MSH and EGT) and ROI production in bacteria [70], and we demonstrated that Cds1-generated H_2_S plays a role in maintaining *Mtb* redox homeostasis. Our data show that catalase in growth media influences ROI production and that H_2_S can function as an oxidant or reductant depending on the environmental conditions. In this regard, careful consideration should be given to experimental design and subsequent conclusions, since exogenous catalase in 7H9 medium is widely known to influence INH drug susceptibility and is routinely excluded from medium in INH susceptibility studies. It has been suggested that 3MST-derived H_2_S protects *E. coli* against oxidative stress via H_2_S-mediated sequestration of Fe^2+^ [14]. In *Mtb*, exogenous H_2_S upregulates key members of the copper regulon, suggesting that copper, which has a high affinity for H_2_S, may contribute to increased ROI when exposed to excess H_2_S [22].

The unusual effect of H_2_S on *Mtb* respiration and oxidative stress, as demonstrated in this study as well as previous studies on how H_2_S impacts antibiotic resistance [15], guided us toward considering a role for H_2_S in the cellular response to anti-TB drugs, such as CFZ, which kills *Mtb* via ROI production [71]. Indeed, our data demonstrate that H_2_S exacerbates oxidative stress in the presence of CFZ. Not surprisingly, Δ*cds1* cells are more resistant to CFZ, consistent with our data showing that addition of exogenous H_2_S to CFZ-treated cells increases *Mtb* killing. On this basis, we propose that endogenous H_2_S stimulates respiration leading to increased ROI production that synergizes with ROI produced by CFZ, ultimately increasing killing. Whereas previous studies have shown that addition of Cys to *Mtb* cells increases respiration and killing of *Mtb* persisters treated with INH and RIF [49], our study suggests that H_2_S could be the effector molecule in that model. It should be recognized that exogenous Cys also generates H_2_O_2_ to trigger the Fenton reaction, which leads to continuous ^•^OH formation that damages DNA which may exacerbate the effect of INH, particularly in closed vessels where H_2_O_2_ accumulates [50]. Exogenous H_2_S in combination with RIF also increased *Mtb* killing, likely because RIF also generates oxidative stress [72], but this effect was less pronounced. These findings may have important implications for TB therapy, as they suggest that sulfur sources in vivo could influence anti-TB drug efficacy. Similarly, recent studies have shown that exogenous H_2_S with RIF (and other antibiotics) confer hypersensitivity to *Acinetobactor baumannii* [73].

## 5. Conclusions

In this study, we showed that pathogenic *Mtb* strains produce H_2_S mainly through Rv3684/Cds1 to regulate energy metabolism and ameliorate cysteine toxicity. Our findings represent a significant conceptual advance that may broadly impact the TB field, especially since H_2_S production by *Mtb* is a previously overlooked confounding factor in routine TB experimentation. Our findings present a basis for understanding how *Mtb*-derived H_2_S regulates *Mtb* OXPHOS and glycolysis, redox homeostasis, and anti-TB drug susceptibility. These findings may also contribute to original virulence paradigms whereby host- and *Mtb*-generated H_2_S subverts host immunity. Lastly, we anticipate that our findings will contribute to a fresh understanding of phenotypic variation in clinical strains of *Mtb* as well as novel diagnostics based on H_2_S production.

## Figures and Tables

**Figure 1 antioxidants-10-01285-f001:**
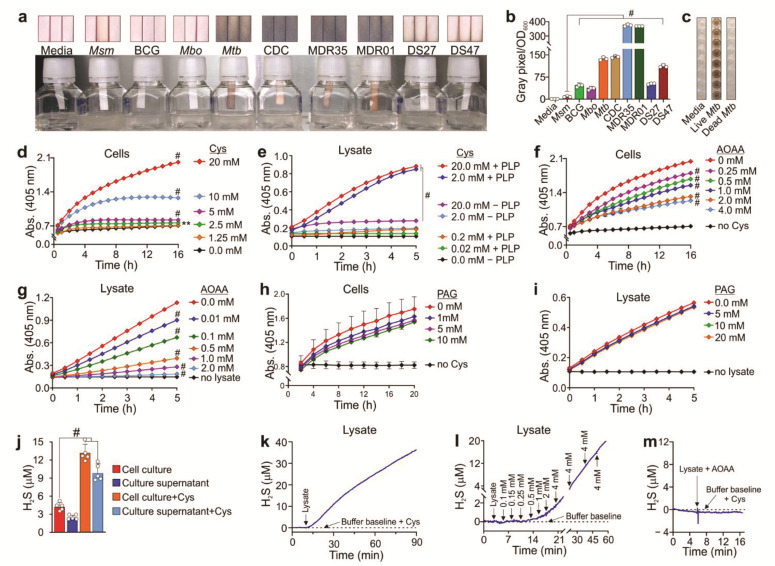
Mycobacterial species produce H_2_S. (**a**) Detection of H_2_S in the headspace of cultures of mycobacterial species using lead acetate strips. *Msm*: *M. smegmatis*; BCG: *M. bovis* BCG; *Mbo*: *M. bovis*; *Mtb*: *Mtb* H37Rv; CDC: *Mtb* CDC1551; MDR: multi-drug resistant; DS: drug-sensitive clinical strains of *Mtb*. Note that strips shown, ((**a**) top insert) were scanned after 48 h of incubation. The inkwell bottles shown are representative of an independent experiment after 72 h of incubation. (**b**) Estimation of H_2_S production by quantifying lead sulfide staining ((**a**) top insert) using densitometric analysis. Data normalized to the optical density (OD_600_) of each culture, (*n* = 3). (**c**) Microplate-based BC assay showing H_2_S production by live and heat-killed *Mtb* H37Rv. Time course measurement of H_2_S production using a BC assay for (**d**) intact *Mtb* H37Rv cells, (*n* = 5–8) and (**e**) lysates with different concentrations of Cys, (*n* = 3–4). BC assay of (**f**) intact *Mtb* H37Rv cells (*n* = 3–5) and (**g**) *Mtb* H37Rv lysates in the presence of AOAA showing reduced H_2_S production, (*n* = 4). No significant inhibition of H_2_S production for (**h**) intact *Mtb* H37Rv cells (*n* = 3–4) and (**i**) lysates was demonstrated in the presence of PAG, (*n* = 4). A Unisense A/S H_2_S microsensor was used to measure the H_2_S concentration in (**j**) *Mtb* H37Rv cultures and bacteria-free supernatants, (*n* = 4). Microsensor measurements of real-time H_2_S production in *Mtb* H37Rv lysates in assay buffer (0.2 M triethanolamine–HCl, pH 8.0; 10 µM PLP, 10 mM EDTA) with (**k**) 20 mM L-Cys, (**l**) stepwise addition of L-Cys (arrows indicate the amount of Cys [mM] added at the time point) and (**m**) 20 mM L-Cys followed by the addition of *Mtb* lysate preincubated with AOAA (4 mM). Representative experiments are shown; each experiment was repeated at least twice. Data represent the mean ± SD from the indicated *n* (number of replicates per data set) except for (**k**,**l**,**m**) which show representative real-time measurements for one sample with 2–3 independent repeats. All *p*-values are relative to untreated controls or as indicated. Statistical analyses were performed using GraphPad Prism 8.4.3. One-way ANOVA with Dunnett’s multiple comparisons test was used to determine statistical significance. # *p <* 0.0001.

**Figure 2 antioxidants-10-01285-f002:**
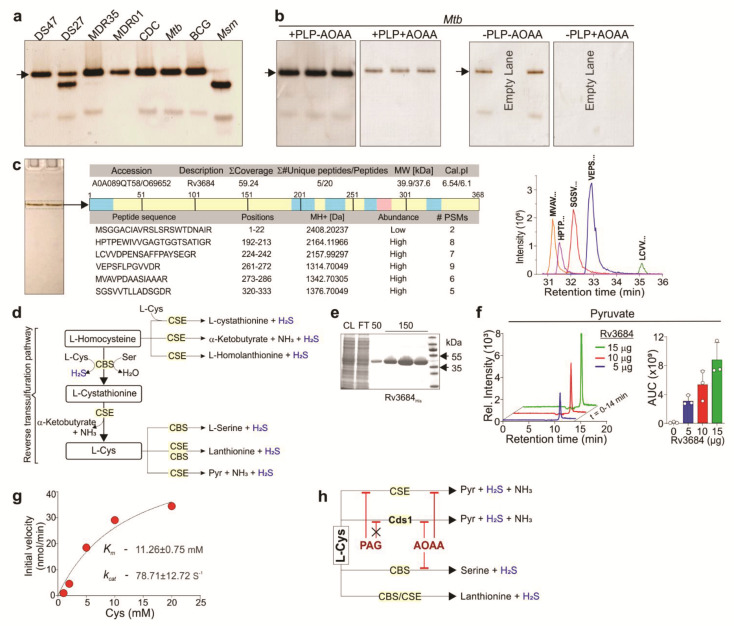
Identification of H_2_S-producing enzymes in *Mtb*. (**a**) Equal quantities of mycobacterial lysates were resolved on a native polyacrylamide gel (PAGE) and assayed for H_2_S production by an in-gel BC assay. The arrow indicates the predominant H_2_S-producing enzyme, upper band, for most mycobacterial species except for *Msm.* (**b**) The effect of PLP and AOAA on the production of H_2_S in *Mtb* H37Rv lysate using an in-gel BC assay. All lanes were loaded with equal amounts of lysate (*n* = 2–3). (**c**) Identification of Rv3684 from *Mtb* lysate resolved by native PAGE. Trypsin-digested peptide fragments of Rv3684 were identified using LC-MS/MS. The locations of the observed peptide fragments (blue and pink regions within the full-length peptide map) and their amino acid sequences are shown. PSM: peptide spectrum matches; MH+ (Da): protonated, monoisotopic mass of the peptide. Retention time of each LC-MS/MS identified peptide of Rv3684 are shown (**right** panel, (**c**)). (**d**) Schematic showing H_2_S-generating enzymes and their reactions in the reverse transsulfuration pathway. (**e**) SDS-PAGE analysis of eluted fractions of purified His-tagged Rv3684 expressed in *E. coli*. (**f**) LC-MS/MS identification and quantification of pyruvate from Rv3684 using L-Cys (10 mM) as a substrate, (*n* = 3). (**g**) Michaelis–Menten plot of the initial reaction velocity as a function of Cys concentration used to determine the K_m_ and k_cat_ of purified Rv3684. H_2_S-producing activity of purified Cds1 (Rv3684) using a BC assay, (*n* = 4). (**h**) Proposed catalytic activity of Cds1 using Cys as a substrate. Representative experiments are shown; each experiment was repeated at least twice. Data represent the mean ± SD from the indicated *n* (number of replicates per data set).

**Figure 3 antioxidants-10-01285-f003:**
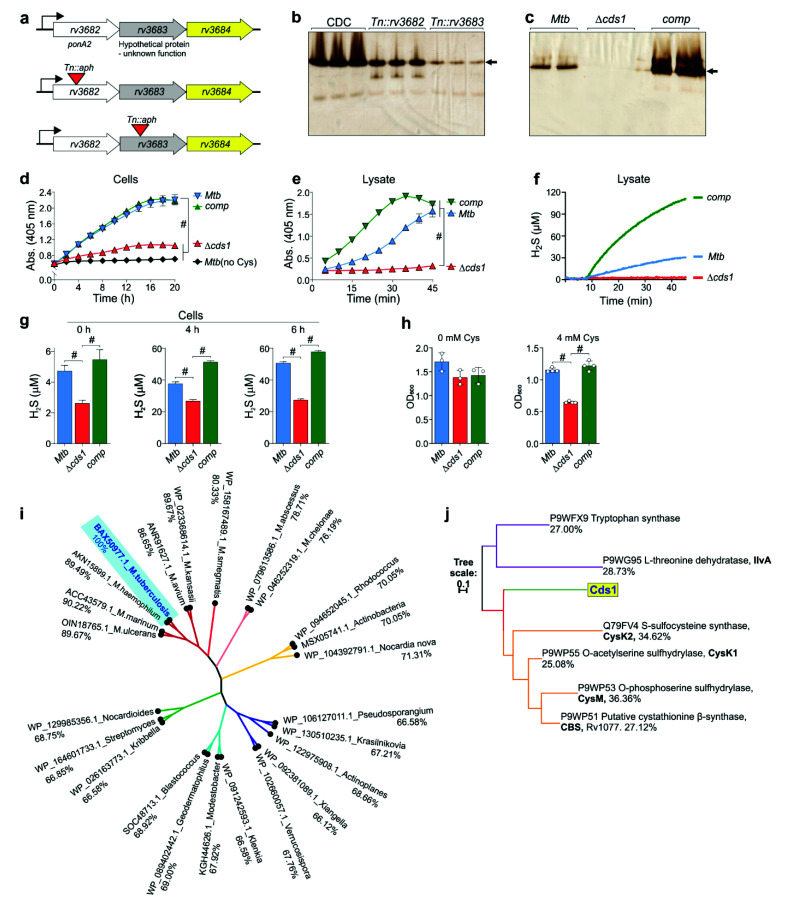
Deletion of *rv3684* in *Mtb* reduces H_2_S production. (**a**) The proposed operonic location of *rv3684*. Bacterial lysates resolved on native polyacrylamide gels (PAGE) were assayed for H_2_S production using an in-gel BC assay in (**b**) *Mtb* CDC1551, CDC *Tn::rv3682*, and CDC *Tn::rv3683* (*n* = 3) and (**c**) *Mtb* H37Rv, ∆*cds1*, and *comp*, (*n* = 2). The arrow indicates the Cds1 band. (**d**) H_2_S production in intact *Mtb* H37Rv, ∆*cds1,* and *comp* cells (*n* = 8) and (**e**) lysates (*n* = 3–4) using the BC method (in the presence of 10 μM PLP and 20 mM L-Cys). (**f**) H_2_S production in *Mtb* H37Rv, ∆*cds1*, and *comp* lysates and (**g**) cells at 0, 4, and 6 h using the Unisense amperometric microsensor. (**h**) Growth (OD_600_) of *Mtb* H37Rv, ∆*cds1*, and *comp* cultures after 8 days in 0 and 4 mM L-Cys, (*n* = 3–4). (**i**) Phylogenetic tree of Cds1 (Rv3684) homologs in different *Mycobacterium* sp. and other bacteria (NCBI protein accession number) and >65% amino acid identity with Cds1. (**j**) Putative sulfur metabolism proteins in *Mtb* H37Rv (NCBI protein accession number and the percent identity with Cds1 provided). Representative experiments are shown in each panel, each experiment was repeated at least twice. Data represent the mean ± SD or ± SEM (panel (**e**)) from the indicated *n* (number of replicates per data set). (**f**) The microsensor reading for a buffer baseline for a few min followed by addition of the lysate to initiate the reaction. (**g**) At least 10–20 amperometric measurements were taken for each strain and condition once the Unisense microsensor readings stabilized. Statistical analyses were performed using GraphPad Prism 8.4.3. One-way ANOVA with Dunnett’s multiple comparisons test was used to determine statistical significance. # *p* < 0.0001.

**Figure 4 antioxidants-10-01285-f004:**
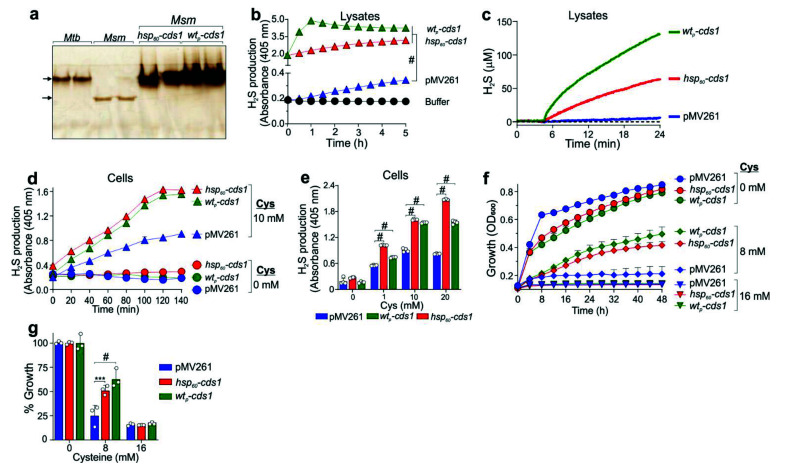
Conferring H_2_S-producing activity to *M. smegmatis*. (**a**) Equal quantities of bacterial lysates from *Mtb* H37Rv, *Msm*, and *cds1*-expressing *Msm–wt_p_–*c*ds1* and *Msm–hsp_60_–*c*ds1* cells were resolved on native polyacrylamide gels (PAGE) and assayed for H_2_S production using an in-gel BC assay, (*n* = 2). (**b**) BC assay of H_2_S production in *Msm* lysates. pMV261-*Msm* transformed with an empty pMV261 vector (*Msm*-pMV261) and a buffer–lysate suspension buffer and BC solution, (*n* = 4). (**c**) H_2_S production in *Msm* lysates using the Unisense amperometric microsensor. The microsensor readings were allowed to measure a buffer baseline for 5 min followed by addition of the lysate to initiate the reaction. (**d**) Time course of H_2_S production in intact *Msm* cells in the presence of 0 and 10 mM L-Cys in the BC assay, (*n* = 4). (**e**) H_2_S production in intact *Msm* cells in the presence of 0, 1, 10, and 20 mM L-Cys after 140 min in the BC assay, (*n* = 4). (**f**) Growth kinetics (OD_600_) of *Msm* strains in Cys-containing media, (*n* = 3). (**g**) Relative percent growth of *Msm* strains in L-Cys-containing media after 48 h, (*n* = 3). Representative experiments are shown; each experiment was repeated at least twice. Data represent the mean ± SD or ± SEM (**d**,**f**) from the indicated *n* (number of replicates per data set). Statistical analysis was performed using GraphPad Prism 8.4.3. One-way ANOVA (**b**) and two-way ANOVA (**d**,**f**), and Dunnett’s multiple comparisons test were used to determine statistical significance. *** *p* < 0.001, # *p* < 0.0001.

**Figure 5 antioxidants-10-01285-f005:**
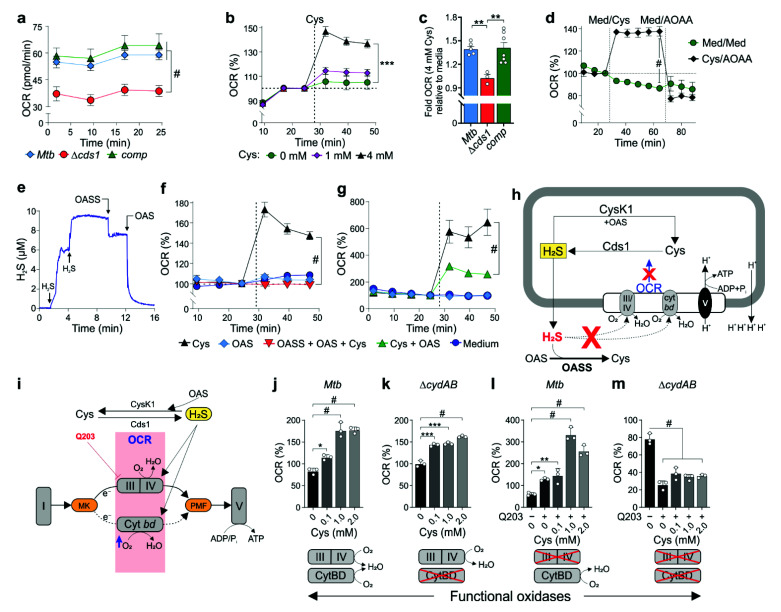
Endogenous H_2_S stimulates respiration in *Mtb*. *Mtb* respiration was measured using an Agilent Seahorse XFe96 Analyzer. (**a**) An OCR profile showing the basal respiration of *Mtb* strains (*n* = 5–7), (**b**) %OCR of *Mtb* upon addition of Cys (*n* = 3–5), (**c**) fold OCR of *Mtb* strains relative to media control in the presence of 4 mM Cys (*n* = 3–7), (**d**) %OCR of *Mtb* with sequential injection of Cys and AOAA (1 mM) or media (Med) as a control (*n* = 3), and (**e**) measurement of H_2_S concentration in an OASS enzymatic activity assay using the Unisense H_2_S microsensor. H_2_S levels rapidly diminished after addition of OASS and substrate *O*-acetyl-L-serine (OAS). (**f**) %OCR of *Mtb* after injecting either Cys or OAS or Cys, OAS, and OASS, (*n* = 3–4). (**g**) %OCR of *Mtb* after injecting either Cys or Cys and OAS, (*n* = 3–4). (**h**) Model showing how H_2_S stimulates respiration via both oxidases (dotted-line arrows) and how respiration was reduced by the exogenous addition of the H_2_S-consuming enzyme OASS and endogenous CysK1. Red “X”: inhibition: blue arrow: increased OCR. (**i**) Model showing electron flow through Complex I to the menaquinone pool (MK) and then through Complex III/IV (cytochrome *bc_1_/aa_3_*) or re-routing of electrons through cytochrome *bd* if Complex III/IV was inhibited by Q203. This contributes to the proton-motive force that powers ATP synthesis by Complex V (ATP synthase). The OCR profiles of WT *Mtb* (**j**,**l**) or Δ*cydAB* cells (**k**,**m**) exposed to Cys or Cys and Q203 (300 × MIC_50_), (*n* = 3). Representative experiments are shown; each experiment was repeated at least twice. Data represent the mean ± SEM or ± SD (**a**) from the indicated *n* (number of replicates per data set). Statistical analysis was performed using GraphPad Prism 8.4.3. One-way ANOVA with Dunnett’s multiple comparisons test (unpaired *t*-test for panel (**d**)) was used to determine statistical significance. * *p* < 0.05, ** *p* < 0.01, *** *p* < 0.001, # *p* < 0.0001.

**Figure 6 antioxidants-10-01285-f006:**
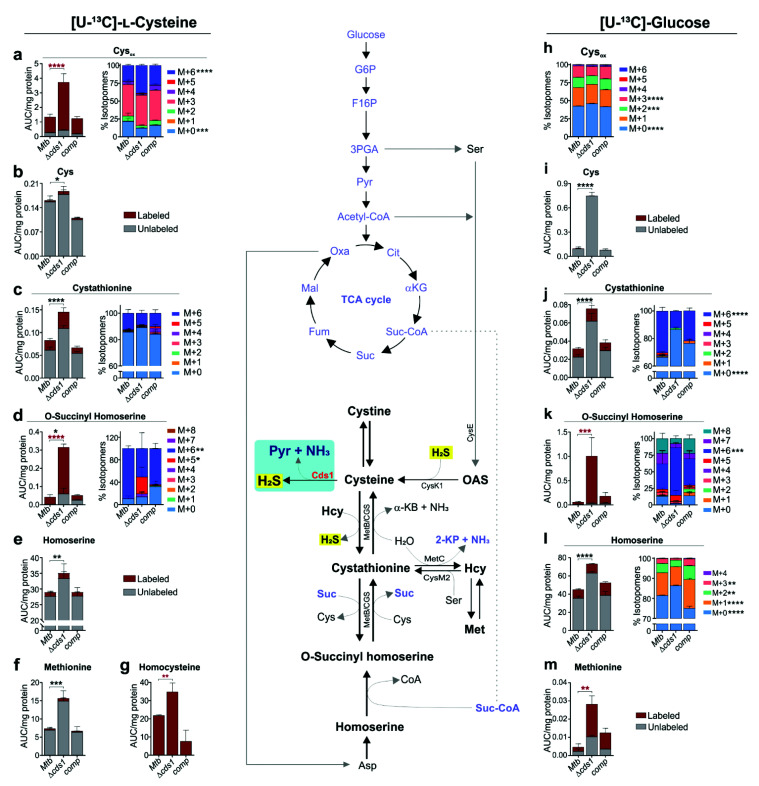
Cds1 regulates sulfur metabolism in *Mtb*. Schematic (center) showing metabolites in the transsulfuration pathway (black) linked with the glycolysis and TCA cycle (blue). LC-MS/MS analysis of metabolites in *Mtb* strains cultured in 7H9 medium with [U-^13^C]-Cys (100 µM) (**a**–**g**) or [U-^13^C]-glucose (0.2%) (**h**–**m**). Representative experiments are shown; each experiment was repeated at least twice. Data represent the mean ± SEM for *n* = 3–5 biological replicates. Statistical analysis was performed using GraphPad Prism 8.4.3. Two-way ANOVA with Dunnett’s multiple comparisons test was used to determine statistical significance. * *p* < 0.05, ** *p* < 0.01, *** *p* < 0.001, and **** *p* < 0.0001.

**Figure 7 antioxidants-10-01285-f007:**
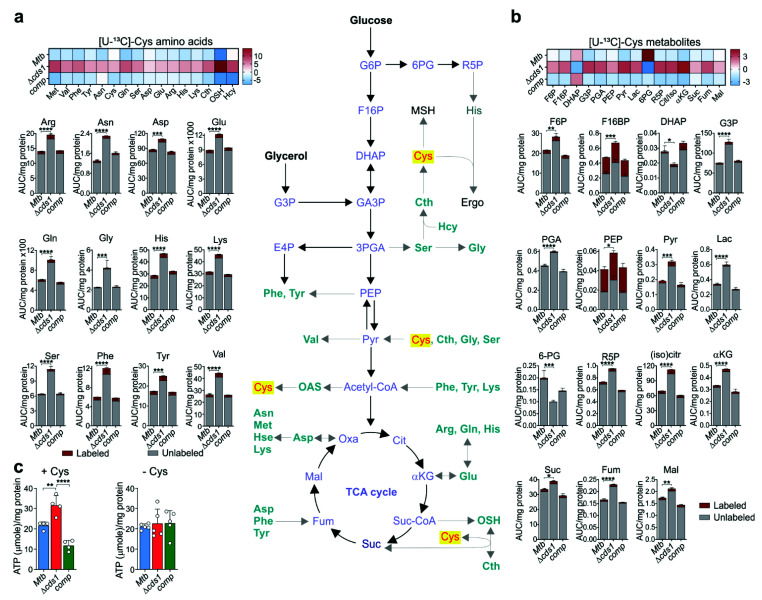
Cds1 regulates central metabolism in *Mtb*. Cds1 in *Mtb* regulates amino acids (turquoise), Cys (red), and glycolysis and TCA cycle metabolites (blue). LC-MS/MS analysis of (**a**) amino acids, (**b**) glycolysis, the pentose phosphate pathway, and the TCA cycle metabolites is indicated by heat maps (top) and total abundance. *Mtb* strains were cultured in 7H9 medium with [U-^13^C]-Cys (100 µM). (**c**) ATP levels measured in *Mtb* strains with (+) and without (−) 100 mM Cys. Representative experiments are shown; each experiment was repeated at least twice. Data represent the mean ± SEM for *n* = 3–5 biological replicates. Statistical analysis was performed using GraphPad Prism 8.4.3. Two-way ANOVA (**a**,**b**) or one-way ANOVA (**c**), Dunnett’s multiple comparisons test were used to determine statistical significance. * *p* < 0.05, ** *p* < 0.01, *** *p* < 0.001, **** *p* < 0.0001.

**Figure 8 antioxidants-10-01285-f008:**
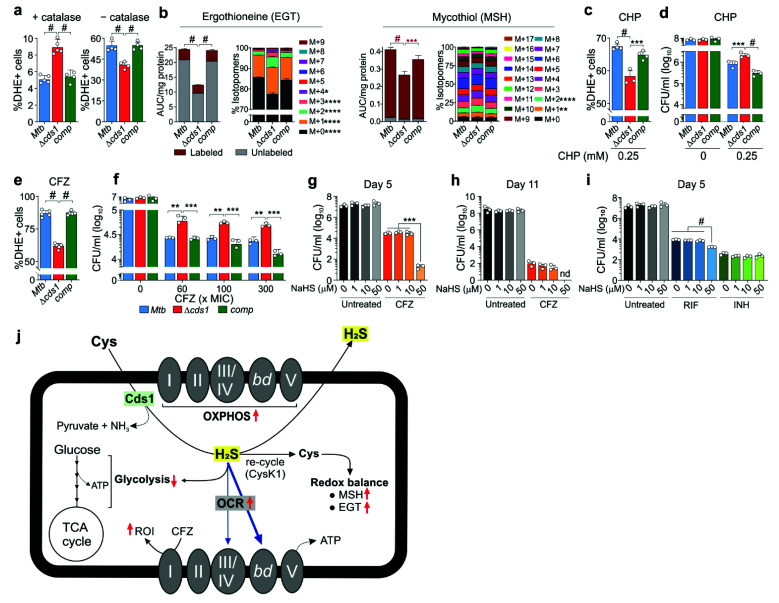
Endogenous H_2_S in *Mtb* exacerbates the effect of oxidative stress and increases susceptibility to CFZ: (**a**) Intracellular ROI levels shown as the percent of DHE-positive *Mtb* cultured in 7H9 media supplemented with or without catalase, (*n* = 4–5). (**b**) LC-MS/MS analysis showing total abundance and carbon isotopologue distribution of EGT and MSH in *Mtb* strains cultured in [U-^13^C]-glucose, (*n* = 3–5). (**c**) Intracellular ROI levels after exposure to 0.25 mM cumene hydroperoxide (CHP) for 16 h, (*n* = 4). (**d**) Survival of bacilli after exposure to 0.25 mM CHP for 16 h, (*n* = 3). Bacilli were treated with clofazimine (CFZ) at 60× MIC and observed for (**e**) ROI levels after 24 h of treatment, (*n* = 5) and (**f**) survival after 8 days of treatment, (*n* = 3). CFU of *Mtb* after exposure to different amounts of NaHS with or without 60× MIC of CFZ for 5 days (**g**) or 11 days (**h**), (*n* = 3). (**i**) Survival of *Mtb* after exposure to different amounts of NaHS with or without RIF (60× MIC) or INH (60× MIC) for 5 days, (*n* = 3). (**j**) Diagram showing endogenous H_2_S production in *Mtb* leading to increased respiration (OXPHOS) and suppression of glycolysis. Indicated, also, is recycling of H_2_S to generate Cys, which regulates redox balance via MSH and EGT. Red arrows pointing up: increase; down: decrease. Thick/thin blue arrows: more/less pronounced effects, respectively. Representative experiments are shown; each experiment was repeated at least twice. Data represent the mean ± SD (mean ± SEM for panel (**b**)) from the indicated *n* (number of biological replicates). Statistical analysis and data presentation performed using GraphPad Prism 8.4.3. One-way ANOVA with Tukey’s (**a**,**e**,**i**) or Dunnett’s (**c**,**g**,**h**) multiple comparisons test were used to determine statistical significance. Two-way ANOVA with Tukey’s multiple comparisons test was used to determine statistical significance for (**b**,**d**,**f**). * *p* < 0.05, ** *p* < 0.01, *** *p* < 0.001, **** *p* < 0.0001, # *p* < 0.0001; nd = not detected.

## Data Availability

Proteomics data were deposited at the ProteomeXchange Consortium via the PRIDE partner repository with the data set identifier PXD022969.

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
