# Peer review of "Mycobacterium tuberculosis H2S Functions as a Sink to Modulate Central Metabolism, Bioenergetics, and Drug Susceptibility"

_antioxidants, 2021, doi:10.3390/antiox10081285_

Round 1

Reviewer 1 Report

In this paper entitled "Mycobacterium tuberculosis H2S functions as a sink to modulate central metabolism, bioenergetics, and drug susceptibility " the authors identified the enzyme responsible for generating the majority of H2S in M. tuberculosis and have characterised further the implication of the function of this enzyme. I enjoyed reading this paper and the experimental design, set up, transition between experiments and conclusions seem logical.

I would only point out to a couple of things I would like to see

1) One experiments that I would have liked to see is the complementation with the enzyme and a (cysteine to serine?) mutant of the enzyme that renders it non-functional, to see the effect on H2S production in a specific way.

2) Do the levels of expression of Cds1 change over time? Perhaps between Day5 vs Day11 shown in Fig 8?

3) There are no markers in any of the gels with the assays. Perhaps it would be best if the molecular sizes are indicated.

Reviewer 2 Report

In the article: “Mycobacterium tuberculosis H2S functions as a sink to modulate central metabolism, bioenergetics, and drug susceptibility”, the Authors measured H2S production in M. tuberculosis (Mtb) by using three different methods and reported that Rv3684 (Cds1) produced H2S using cysteine as a substrate in Mtb.  Further, they also studied the impacts of H2S generated by Cds1 on the mitochondrial respiration and sulfur and central metabolism in Mtb.  Nonetheless, the paper is well written, and the experimental part is well-done, and the work should be published following the following 'major' considerations.

Major comments

1) The authors used three different methods to detect H2S production in Mtb.  However, the BC assay, was mainly employed for the H2S production in this study, may detect not only H2S but also other sulfides such as reactive sulfur species which is recently found to be endogenously produced in various organisms including E. Coli. and Salmonella Typhimurium LT2 [Ref: PMID 29079736 and 30197193].  Further, reactive sulfur species can easily degrade into H2S during sample preparation procedures, especially under alkaline conditions [Ref: PMID 29909607 and 30634125], leading to a misinterpretation of the endogenous H2S production.  Therefore, the authors should investigate whether reactive sulfur species contribute to H2S production in Mtb.  To specifically detect reactive sulfur species, a fluorescent assay with specific probes such as SSP4 developed by Prof. Ming Xian or a mass spec analysis is recommended.

2) The authors identified Cds1 as a H2S-producing enzyme in Mtb by a mass spec analysis (Fig. 2c).  However, the sequence coverage of Cds1 protein was too low.  Further, the kinetic analysis with recombinant Cds1 protein revealed that the Km value for cysteine with Cds1 was about 11 mM (Fig. 2g).  Can Cds1 efficiently produce H2S using cysteine in Mtb?  What is the cysteine concentration in Mtb?  Considering the kinetic parameter, it is doubtful that Cds1 can directly contribute to the H2S production as a H2S-producing enzyme in Mtb.  It has recently reported that bacterial cysteine-tRNA ligase can mediate desulfuration of cysteine to produce reactive sulfur species and H2S and the enzymatic activity and the enzymatic activity increases in a PLP-dependent manner [Ref: PMID 29079736].  They also demonstrated that the Km value for cystine with recombinant E. Coli. cysteine-tRNA ligase was around 7 μM.  Since cysteine-tRNA ligase is ubiquitously expressed in not only procaryotes but also eucaryotes, cysteine-tRNA ligase may contributes to the H2S production in Mtb.  The authors should examine or discuss the possibility.

Minor comments

1) It is difficult for this Reviewer to read Figure 3g because of the poor resolution.  The author should provide the Figure with a high resolution.

Round 2

Reviewer 2 Report

  1. The authors states that the current literature indicates that the BC assay specifically reacts with H2S. There is, however, no evidence to examine the specificity of the BC assay in the current study. There is a possibility that the BC assay detect other sulfide-containing molecules such as persulfides rather than H2S. Therefore, to precisely conclude the contribution of cds to the H2S production in Mtb, the authors need to measure H2S production in the Mtb rv1077/cbs deletion strain (Δcbs) and comp cells using a H2S-specific detection technique such as the Unisense H2S probe sensor.

  1. The authors should measure or at least discuss the endogenous concentration of cysteine and other sulfide-containing molecules in Mtb and the substrate specificity in the Discussion section.

Author Response

REVIEWER COMMENT #1.

The authors states that the current literature indicates that the BC assay specifically reacts with H2S. There is, however, no evidence to examine the specificity of the BC assay in the current study.

Response: We agree with the Reviewer.  The basis of the Bi3+ assay is the formation of the highly insoluble (Ksp = 1.82 x 10-99) black salt precipitate bismuth sulfide (Bi2S3).  This will not form with covalent compounds containing sulfur (including sulfane) since a salt (complete electron transfer from the sulfur to the bismuth) does not form.  A statement to this effect is included in lines 476-478 in the revised manuscript.

There is a possibility that the BC assay detect other sulfide-containing molecules such as persulfides rather than H2S. Therefore, to precisely conclude the contribution of cds to the H2S production in Mtb, the authors need to measure H2S production in the Mtb rv1077/cbs deletion strain (Δcbs) and comp cells using a H2S-specific detection technique such as the Unisense H2S probe sensor.   

Response: We thank the Reviewer for these comments and have perform all the recommended experiments (as indicated below): 

  • We have used the Unisense H2S microsensor and examined H2S production in wild-type Mtb, the Δcds1 mutant and corresponding complemented whole cells, and lysates. This has been discussed in lines 621-622, 665-666, and 671-673.  These data resulted in two new figure panels (Figure 3f, g) in the revised manuscript.  The Materials and Methods section has also been updated (lines 183-192) in the revised manuscript.
  • We have also used the Unisense H2S microsensor to examine H2S production in Msm lysates, which has been described in lines 691-692, and lines 697-699 and resulted in a new figure panel, Figure 4c in the revised manuscript.
  • Overall, our Unisense microsensor findings (whole cells and lysates) are consistent with our BC assay.

We thank the Reviewer for help improving the scientific rigor of our manuscript.

REVIEWER COMMENT #2

The authors should measure or at least discuss the endogenous concentration of cysteine and other sulfide-containing molecules in Mtb and the substrate specificity in the Discussion section.

Response: We thank the Reviewer for the comments.  We have discussed the sulfur concentration in bacteria in lines 1051-1054 (highlighted in yellow).  We prefer not to discuss other sulfide-containing molecules as there are too many, and their molar concentrations are unknown, which may be distracting to the readers.  Although we feel that the substrate specificity of Cds1 is the focus of independent study, we provide a limited discussion (lines 1075-1077; highlighted yellow).

Round 3

Reviewer 2 Report

The authors reflected to all the questions and  provided new experimental data to improve the manuscript.